# A Walk in the Memory, from the First Functional Approach up to Its Regulatory Role of Mitochondrial Bioenergetic Flow in Health and Disease: Focus on the Adenine Nucleotide Translocator

**DOI:** 10.3390/ijms22084164

**Published:** 2021-04-17

**Authors:** Anna Atlante, Daniela Valenti

**Affiliations:** Institute of Biomembranes, Bioenergetics and Molecular Biotechnologies (IBIOM)-CNR, Via G. Amendola 122/O, 70126 Bari, Italy

**Keywords:** adenine nucleotide translocator, mitochondria, ATP detecting system, transport, physiological role, disease

## Abstract

The mitochondrial adenine nucleotide translocator (ANT) plays the fundamental role of gatekeeper of cellular energy flow, carrying out the reversible exchange of ADP for ATP across the inner mitochondrial membrane. ADP enters the mitochondria where, through the oxidative phosphorylation process, it is the substrate of Fo-F1 ATP synthase, producing ATP that is dispatched from the mitochondrion to the cytoplasm of the host cell, where it can be used as energy currency for the metabolic needs of the cell that require energy. Long ago, we performed a method that allowed us to monitor the activity of ANT by continuously detecting the ATP gradually produced inside the mitochondria and exported in the extramitochondrial phase in exchange with externally added ADP, under conditions quite close to a physiological state, i.e., when oxidative phosphorylation takes place. More than 30 years after the development of the method, here we aim to put the spotlight on it and to emphasize its versatile applicability in the most varied pathophysiological conditions, reviewing all the studies, in which we were able to observe what really happened in the cell thanks to the use of the “ATP detecting system” allowing the functional activity of the ANT-mediated ADP/ATP exchange to be measured.

## 1. Introduction

Thousands of reviews on the adenine nucleotide translocator (ANT) are available on PubMed^®^, the database (continuously updated) comprising more than 30 million citations for biomedical literature from MEDLINE, life science journals, and online books from the 1950s to today.

In particular, in the month of February 2021—the period in which we wrote this review—by typing as the keyword “adenine nucleotide translocator,” more than 18,000 original articles were traced by the system. We tried to narrow down the search by associating two other keys: “regulation” and “function.” The number of results obtained was greatly reduced, but it was always in the thousands. Of these, about 1500 were reviews.

All this is to say that this study of ours does not pretend to present yet another review on ANT to the scientific world—after all we do not have the necessary skills to embrace the subject from the point of view of genomics, proteomics, genetics, and molecular and cellular biology—but rather to tell how from our first entry in a biochemistry laboratory 35–40 years ago up to the present day, we have dealt with how this mitochondrial transport protein works.

To be precise, ours is a study dedicated to measuring the rate of appearance of ATP outside the mitochondria, under conditions close to a physiological environment that reflected what actually happens in the cell. Our main aim is to put the spotlight on the validity and versatility of this measurement method, which has not received adequate consideration, in our opinion, although it has been applied in the most varied pathophysiological conditions we have studied, as we will see later (Section 3).

We were still students when we collaborated in the tuning of the method that allows for the measuring of the ADP/ATP exchange by monitoring the extramitochondrial efflux of ATP gradually produced inside mitochondria for oxidative phosphorylation (OXPHOS) as a result of ADP addition and uptake into mitochondria. Even today we meet ANT in our study/research path in a great variety of diseases in which it is involved due to its pivotal role in cellular metabolism. Not only will we see in this roundup of research what the effect of laser light is on ANT activity and how it reacts following drug photosensitization of isolated mitochondria, but also how the antiviral drug 3′-Azido-3′-deoxythymidine (AZT) affects it, and even how its impairment leads to mitochondrial energy deficit in Down syndrome (DS), as well as its involvement in Alzheimer disease (AD) progression, in which its function intertwines with those of the two AD proteins, i.e., β-amyloid and TAU, and so on.

However, in order to facilitate reading, also addressing readers who deal with something else, but who are interested in knowing how the energy produced by mitochondria is transferred and made available in the cytosol, first we will briefly present mitochondria—recalling our first approach with them when we were students—and describe their functional mechanisms as well as the salient features of ANT, focusing essentially on those that will be taken into consideration in the subsequent paragraphs, postponing the reading to strictly specific studies. This will allow the reader to understand the experimental strategy adopted by the method used to monitor the transfer of energy in the form of ATP from the inside of the mitochondria to the outside, a method that, since its development more than 30 years ago, is still at the forefront, when the study of the cell state from a bioenergetic point of view is addressed under different pathophysiological conditions.

## 2. Mitochondria: Not Just an Energy Production Site

For more than a century, biologists have looked at mitochondria as bean-shaped organelles floating in the cytoplasm and working tirelessly and without respite to provide the energy required for life. However, that simple picture of mitochondria soon proved incredibly incomplete, especially since mitochondria do not have a stable shape—they stretch, shrink, blend, and divide all the time—and energy production is not their only role.

Before getting into the essential core topic of this review, we will provide some useful information about mitochondria in order to allow newbies to understand it.

### 2.1. Two Membranes and Two Spaces with Specific Functions

Mitochondria possess two highly specialized membranes that together create two separate mitochondrial compartments, namely, the intermembrane space and the matrix.

The inner mitochondrial membrane (IMM), unlike the external one that envelopes the organelle, is sub-divided into two compartments, the inner boundary membrane (IBM) and the cristae membranes (CM) [1], with a surface 1.5–2-fold larger than the IBM part [2]. Both parts are connected via so-called cristae junctions that are often slit-like, as electron tomography has revealed [3]. Although by and large they constitute a continuous membrane, the sub-compartments achieve different tasks and thus have a distinct protein composition [2]. For example, the IBM is the site of protein import and thus displays a reliable presence of transport proteins, called the translocase of the inner membrane, TIM [4]. It also hosts other carriers and channel proteins. On the contrary, the cristae are the bioenergetic units harboring the five mitochondrial respiratory complexes (mRC), NADH:ubiquinone oxidoredutase (NADH-dehydrogenase, complex I, CI), succinate dehydrogenase (SDH, complex II, CII), Cytochrome *c*-reductase (b*c*1 complex, complex III, CIII), cytochrome *c* oxidase (complex IV, CIV), and ATP synthase (complex V, CV). All together, they form the OXPHOS system, which is not the only process of the cell’s energy metabolism involving mitochondria.

The inner compartment of the matrix is moderately dense and hosts strands of DNA and ribosomes, molecular tools through which the mitochondria are able to code for part of their proteins. The matrix contains the complete catalytic kit of enzymes that take part in terminal oxidative metabolism, such as the Krebs cycle, the beta-oxidation of fatty acids and the urea cycle. The metabolism of amino acids, lipids, cholesterol, steroids, and nucleotides also takes place in this mitochondrial district.

As for the outer mitochondrial membrane (OMM), which is essentially smooth and elastic, it is characterized by a high lipid/protein ratio and contains a channel for anions, whose activity is dependent on the electric potential, the voltage dependent anion channel (VDAC), also called “porin,” together with devices used both for communication with other cellular structures and for the recognition and import of mitochondrial proteins encoded by the nucleus [5]. It seems that VDAC controls the passage of metabolites between the cytosol and mitochondrial intermembrane space via the modulation of different open and closed states of the protein [6,7]. In this regard, we refer the reader to Section 4.2.4.1, in which VDAC1 activity decreases in concomitance of its physical interaction with hexokinase (HK) [8]. In addition, through its location in the OMM and association with various ligands and proteins, it serves as a convergence point for a variety of cell survival and death signals, linking energy, redox, and signaling pathways in mitochondria and other cellular compartments. Regarding this, it is interesting to note, for example, that HK and glycerol kinase interact with the porins of the external surface of the OMM [9] in such a way as to have preferential access to the ATP generated in the mitochondria; in addition, the binding of HK with mitochondria has been considered the basis for energy metabolism in cell proliferation [10].

The regulation operated by the VDAC causes a concentration of substrate different (or higher) than that of the extramitochondrial phase similar to cytosol, to establish itself in the intermembrane space due to the phenomenon of micro-compartmentation. Although we will return to this topic soon, now just a digression about the concept of micro-compartmentation presupposes that (i) in the mitochondria microenvironments are created with a composition of substrates different from that of the surrounding environment and that (ii) the mutual interaction of proteins can lead to a sort of “channeling” in which the products become reagents for enzymes that are physically “close/bound together,” more or less a situation similar to mRC supercomplexes (see Section 3.6.1).

### 2.2. As Students in a Bioenergetics Laboratory

We first heard the term “bioenergetics” when, as internal students, we entered the renowned Institute of Biological Chemistry at the University of Bari. In our lab everything contributed to creating a bioenergetics atmosphere: The electrons came and went along the mRC, the Gilson oxygraph marked the notches on the paper of the recorder following the addition of respiratory substrates causing oxygen consumption, and discussions of how mitochondria generate energy and how metabolites enter and/or leave the mitochondria and so on were on the agenda. One thing is certain: our research has always covered basic aspects of mitochondrial bioenergetics and, in the last few years, it has ranged in the field of the regulation of energy metabolism in health and disease.

#### 2.2.1. Those Were Times When

Those were years of discovery and great ferment in the scientific world of bioenergetics and isolated mitochondria were the chief playground for the “experts in bioenergetics.” In Bari, Ernesto Quagliariello carried out an intense scientific and organizational activity dotted with international events. For about 20 years prior, in fact, annual meetings on mitochondria—organized by researchers from the laboratory of biochemistry at the University of Bari, in collaboration with their colleagues from other Italian and foreign universities—were held in Bari or in the villages around Bari. These meetings were a tradition. They saw the participation of leading biochemists and molecular biologists, including Albert Lehninger; Sir Hans Krebs, Nobel Prize in Chemistry in 1953; Bill Slater and Britton Chance; Vladi Skulachev, friend and regular visitor to Bari; and John Walker, Nobel Prize in Chemistry in 1997. It was in one of these meetings that Peter Mitchell presented an articulated speech on chemiosmotic theory, a topic that in 1978 earned him the Nobel Prize in Chemistry. It was also at one of the Bari meetings, where all new aspects of mitochondrial processes were debated, that the term “bioenergetics” was launched, having becoming more popular after Lehninger’s little book “Bioenergetics” was first published in 1964.

Among the great scholars hosted, we absolutely cannot forget the German biochemist Martin Klingenberg, who carried out research mainly on mitochondria, in particular on the ADP–ATP transporter.

It is worth mentioning that Martin Klingenberg and his co-workers were among the first to fully recognize the implications of Mitchell’s work on metabolite transport across the IMM. In further pioneering studies, his laboratory staff purified the ADP/ATP carrier [11] and reconstituted that protein into proteoliposomes, thus allowing detailed characterization of its transport properties [11,12,13].

At that time, no textbook would have even nearly covered the broad range of topics addressed in “Bari Meetings” lectures. The proceedings of the symposia of the “Bari Meetings”—published in books, whose quotations can be found in the literature—provided year after year a comprehensive and timely record of the biochemical aspects of mitochondrial transport, with great strides towards the understanding of its molecular mechanism. Multiple transport systems were defined in terms of their catalysts, pumping systems and carriers. Therefore, right in the period in which the research on these carrier systems was fervent and competitive, upon our arrival in that laboratory and after becoming familiar with the organelle, we could not help but collaborate in this research that led to the discovery of other new carriers in mitochondria of various origins, also depending on the physiological role of the organ.

#### 2.2.2. Becoming Familiar with the Mitochondrion

At the time of our internship as students—and also afterwards, during the period of learning as trainees, or in any case during doctorate school (for D.V.)—before each experiment, having an exact design goal, the isolation of mitochondria followed by specific tests of functionality, was our daily bread. Thus, we learned not only to treat this precious but delicate organelle with care and skill, but also to grasp the substantial differences between organs, either due to the type of tissue—just consider the heart, whose tissue is tenacious, so the shredding is more cumbersome and the homogenization is repeated longer to allow the walls to break and the mitochondria to escape—or due to the mitochondrial yield. This type of approach was useful for us to gain experience and grow, to be able to face and cope with the challenges we are now involved in.

Without a shadow of a doubt, doing bioenergetics in those days was an art more than a science, as we did not have the highly elaborate and refined techniques of today. Using those of that time, a successful preparation of a mitochondrial suspension was an indispensable prerequisite for the “success” of the type of experiments on “mitochondrial transport.” In fact, the measurements to be made, those concerning transport, could not ignore the fact that the mitochondria were “respiring and phosphorylating.” Thus, no type of experiment began without conducting an oxygraph check.

Working with mitochondria functioning from a bioenergetic point of view, i.e., mitochondria capable of (i) respiring, i.e., consuming molecular oxygen, (ii) creating membrane potential useful for OXPHOS and other processes, and (iii) synthesizing ATP has always been our flagship. All three parameters are essential for mitochondria to participate in cellular metabolism due to the presence of transport proteins on the IMM.

Already at that time, the understanding of mitochondrial transport was changing. In fact, the concept that (i) the transporters mediating the uptake/export of substrates across mitochondria and (ii) the enzymes that metabolize them were subject to a sort of micro-compartmentation (see above, Section 2.1), which seemed to favor the occurrence of reactions in which transporters and enzymes were involved, was reaching an ever-increasing consensus.

#### 2.2.3. There Are Those Who Go and There Are Those Who Come: Traffic across the Mitochondrial Membranes

In light of the considerations made above on the role of the mitochondrial membrane in controlling the flow of metabolites and for the presence of the micro-compartmentation occurring in IMS, and also because mitochondrial transport is related to and sometimes regulated by energy metabolism, we consider in this paper the studies on isolated mitochondria or mitochondria present in a cellular homogenate from tissue or cells—both respiring and phosphorylating—and not those, even valuable ones, that have as their object artificial systems consisting of isolated transport proteins and liposomes. This is because the closer the experimental system is to “nature,” the more reliable the conclusions are that research can allow us to draw. We verified this at our own expense.

Thus, in the 1980s, when LaNoue and Schoolwerth [14,15] wrote a sublime work collecting all the evidence of the existence of mitochondrial carriers discovered up to then, we began our research activity, essentially adopting the use of spectroscopic techniques (fluorescence and photometry) with which the reactions could be monitored both inside the mitochondria and in the extramitochondrial phase. To do this, isolated and coupled mitochondria from rat liver, heart and left ventricle, kidney, brain, and from cerebellar granule cells as well as from tumor cells and even from plants (durum wheat, potato tuber, Jerusalem artichoke), yeast (*Saccharomyces cerevisiae*), and others were used in order to investigate new aspects of metabolism and their permeability properties.

As mitochondria represent closed spaces in the cell, for energy metabolism to take place it is necessary that there be metabolite “traffic” through the mitochondrial membrane: some from the cytosol will enter the mitochondria to be degraded for energy purposes and/or to give to other metabolites; others will come out of the mitochondria and become substrates for some reactions in the cytosol. Obviously, the presence of enzymes in specific compartments is crucial and their catalytic activity is a function of known parameters distinctive of enzymatic reactions. Over the years, the discovery of the existence of multiple translocators involved in different metabolic pathways, made in light of the location of the enzymes that participate in them, have given answers to unsolved questions that remained unanswered in the study of energy metabolism and/or in didactic texts. Furthermore, since most of the translocated substrates or the newly synthesized metabolites produced by their intramitochondrial metabolism can be oxidized in the mitochondrial matrix, their transport can be monitored fluorimetrically as a change in the intramitochondrial cofactor red/ox state [16]. Similarly, as an additional tool for studying transport, a variety of “metabolite detecting systems” were developed, consisting of enzyme/s and cofactor/s designed to photometrically ascertain the appearance outside mitochondria of molecules derived from the mitochondrial metabolism of the substrate taken up. Given that the components of the detection system are mostly present in the cytosol and the mitochondrial metabolism is, at least in part, active, we believe that this approach is quite close to the physiological situation.

An important aspect to consider is the fact that each cell has its own specific metabolism with a specific role for its mitochondria. For example, if we consider the urea cycle, it takes place exclusively in the liver and involves enzymes located in two cellular compartments, i.e., carbamoyl phosphate synthetase, ornithine transcarbamylase, and glutamate dehydrogenase are mitochondrial proteins, whereas all the others are present exclusively in the cytosol. Therefore, it is necessary for ornithine (ORN) and glutamate to enter the mitochondria and for citrulline to come out. In the early 1970s, the existence of a carrier for ORN operating in uniport was demonstrated [17]; subsequently, following the “physiological” logic, the existence of another carrier that works in antiport with citrulline was proposed [18].

Since each organ has its own specific enzymatic kit and therefore its own biochemical specificity, the situation is supposed to be different in other mitochondria. However, unexpectedly, following the swelling of kidney mitochondria in ORN solutions, we observed a collapse of absorption at 546 nm, an index of mitochondrial swelling and evidence of the entry of the amino acid in question into mitochondria. There is a carrier for ORN in kidney mitochondria that is absent in the liver, which exchanges ORN with Pi with 1:1 stoichiometry [19,20]. Obviously, this is an electrophoretic exchange process, with the membrane potential being the energy used. In the kidney, the transport of ORN is physiologically correlated to its catabolism in the renal cell where the ornithine aminotransferase enzyme is responsible for its degradation.

In the light of the example reported, it can be deduced that it is not correct to transfer the specific characteristics of a metabolite transport occurring in mitochondria isolated from an organ or tissue to other mitochondria, i.e., of different origins. However, the hypothesis that what happens in the mitochondria of a certain origin may also occur in others of different origin, and therefore with different characteristics, is plausible.

Only some carriers are exempt from this limitation: those that are ubiquitous. Among them, besides the carrier of the Pi, surely the ANT mediating the ADP/ATP exchange is the first to be mentioned, in accordance with the fact that the primary function of mitochondria in the cell is the synthesis of ATP, the molecule working as the energy currency for cells.

Having all this information available and making use of our skills, our goal was to measure ANT activity in conditions very close to physiological ones.

#### 2.2.4. Methods Used to Follow the Mitochondrial Trafficking of Metabolites

The challenge in measuring transport was essentially represented by the method to be used to follow the entry of a metabolite into the mitochondria or its efflux from them.

It is intuitive that the undisputed guarantee of entry was provided by isotopic methods, for which the radioactivity of the marked metabolite was traced: These were the direct methods. In this regard, the old review by Palmieri and Klingenberg, published in *Methods in Enzymology* in 1979 [21], is exhaustive and treated in the smallest detail. These methods allowed us to study the uptake of a radioactive substrate added outside the mitochondria and/or the efflux of a marked substrate (with which the mitochondria had been loaded) induced by the addition of an unlabeled substrate to the mitochondrial suspension. The method itself, apart from being expensive, requires special laboratories with equipment dedicated only to isotope studies. In using this method, it is crucial to ensure that you measure the substrate that has actually entered the mitochondria, distinguishing it from what remains in the medium and/or is possibly non-specifically bound to the mitochondria. This risk was counteracted with the subsequent introduction of the inhibitor stop technique (for refs see [22]). A very important note characterizing this method: the inhibitor used must be impermeable to the mitochondria, act quickly, and not be harmful to the mitochondria.

We were young at that time and two technicians, Nicola and Andrea Cataldo, supported us with the technique. They, more than the laboratory technicians, were real miniature architects: Using Plexiglas, a square, and a chisel they were able to build an ad hoc system that included a Plexiglas tray in which—at 0 °C, i.e., in the presence of water and ice—aluminum blocks with 12 housings for Eppendorf tubes were immersed.

The uptake reaction took place following the simultaneous addition of the radioactive substrate and/or the inhibitor through 12 metal rods on which the drop of solution to be added was placed, supported by a wooden bracket. That was art!

The direct methods were contrasted by the “indirect” ones that allowed us to follow the transport of a metabolite through a reaction associated with the appearance of the same in the mitochondria or in the extramitochondrial phase. The substantial difference between the two methods, apart from the fact that the indirect method is a “continuous” method because the measurement is performed by continuously recording the substrate uptake by absorbance or fluorescence variations, consists of the fact that in the first case, i.e., the direct one, any mitochondrial activity is blocked to prevent metabolism of the substrate entering the mitochondria; in the second case, the indirect one, metabolism is largely possible.

About this very important note, for many years, fumarate was considered a substrate unable to enter mitochondria. Yet fumarase has mitochondrial localization. What was the cause of this inconsistency? Chappell and Haarhoff [23] came to the conclusion that fumarate is a non-penetrant metabolite, as they did not observed an accumulation of radioactivity in the mitochondria. This was caused by the fact that in the matrix, the labeled fumarate is transformed—via fumarase—into malate, which, in turn, exits from mitochondria in exchange with fumarate itself [24], thus preventing its accumulation [22].

Other advantages of the indirect method include the speed of execution of the measurement, low costs, and also the requirement of small amounts of mitochondria (even lower than 0.1 mg protein). Taking advantage of these “advantages,” we were able, in the following years, to miniaturize the working system by carrying out measurements on mitochondria present in a homogenate of cells or tissue.

However, it is imperative that if you want to measure the transport rate using spectrophotometric or fluorimetric techniques, the transport (not the reaction (s) dependent on it) must limit the rate with which the observed parameter changes. In fact, in a typical experiment, the measured parameter can depend on three steps: (i) transport, (ii) an intramitochondrial reaction, and (iii) an extramitochondrial reaction. It is therefore essential that the step that regulates the rate—that is, limits it—is transport.

Sometimes detergent, which breaks down the permeability barrier, causing an increase in the process rate, gives us information that the limiting step is actually the transport through the membrane. However, it is not always possible to apply this ploy. Indeed, sometimes the solubilization of the membrane operated by the detergent causes inactivation of enzymes involved in the process and/or, for example, in the case of measuring the ADP/ATP exchange rate, the collapse of the ΔΨ on which the carrier’s activity depends. Thus, in order to ascertain which step limits the rate of appearance of ATP as synthesized via OXPHOS, i.e., to distinguish between the rate of ADP/ATP exchange via ANT and the rate of ATP synthesis via ATP synthase, a control flux analysis (the control strength criterion) may be applied, as in [25].

According this “control strength criterion” (see [14,15])—based on the sensitivity of the global process to a transport inhibitor—if the reciprocal of the measured rate is plotted as a function of the non-penetrant inhibitor concentration, the resulting Dixon plot extrapolated to zero concentration provides a measure of transport in the absence of an inhibitor. Thus, the coincidence of the experimental point measured in the absence of an inhibitor with the intercept shows that you are measuring the rate of the transport (see [25]). Anyway, it must be ensured that the detecting system activity does not limit the measured rate.

### 2.3. ADP and ATP Move Back and Forth within the Mitochondria

It was said above that mitochondria participate in various cellular processes and that all this is made possible by the different localization of the enzymes, inside or outside mitochondria. The fundamental reason explains the movement of metabolites back and forth within mitochondria where they are needed. It follows that the cytoplasm and the internal compartments of the mitochondria are put into communication through carrier-mediated transport processes.

The discovery in the 1960s of the existence of protein-mediated transport processes in the mitochondria was due to the English biochemist Brian Chappell. His truly groundbreaking work on the permeability properties of isolated mitochondria, performed formerly in Cambridge and later in Bristol, was highly original and widely influential in the field, earning him an international reputation. His group was able to make continuous indirect measurements of the entry of metabolic substrates into isolated mitochondria by observing the osmotic behavior of mitochondria in a solution containing the specific ion. This work was complemented by direct measurements of substrate and ion transport using radiolabeled substrates and also ion-selective electrodes. The result was the discovery and characterization of some eight specific substrate transport systems in the mitochondrial membrane, a concept that was completely new at the time and had great implications for the understanding of cell metabolism and ATP synthesis. The main carrier systems were first characterized in Chappell’s lab, the earliest ones being the phosphate/OH^−^ exchange system (the phosphate transporter) and the ADP/ATP-exchange system, i.e., ANT. Both these two carrier systems were soon recognized by Mitchell as an essential component of the chemiosmotic circuit.

In the first, the enzyme catalyzes the net transfer of H_3_PO_4_ across the membrane in a neutral process: Since the predominant ionic species present at neutral pH is H_2_PO_4_^−^, the mechanism must involve either the exchange of H_2_PO_4_^−^ for OH^−^ or the cotransport of H_2_PO_4_^−^ with H^+^ [26].

In the other, i.e., ANT, which is highly abundant, constituting about 14% of the proteins in the IMM, the carrier mediates the exchange of ATP for ADP. Most importantly, the flows of ATP and ADP are coupled. It is a wasteful exchange from an energy point of view: about one-fourth of the energy produced by the flow of reducing equivalents along the mRC is consumed to regenerate the continuous membrane potential modified by the exchange of adenine nucleotides. In fact, at physiological pH, ADP has three negative charges and ATP four. The exchange reaction is therefore electrogenic and the directionality of transport is imposed by the concentration gradients of the substrates and by the electrical component (ΔΨ) of the proton gradient so that the entry of ADP and exit of ATP are favored. The freshly coined ATP is then dispatched from the mitochondrion to the host cell to maintain high cytosolic ATP concentrations for reactions that require energy [27]. Wherever the cell needs it, energy is released from ATP by stripping it of a phosphate. This leaves behind ADP and phosphate that within the mitochondrion can be recharged to ATP again. What is exquisitely elaborate is how mitochondria put this plan into action.

It was after 1982, i.e., when the first amino acid sequence, that of the ADP/ATP carrier [28], was identified, that it became clear that all carriers belong to the same “family” by virtue of common characteristics of their molecules.

After the first communications on the discovery of the ADP/ATP exchange [22], a full characterization of the ADP/ATP transport in mitochondria was given in two detailed key papers [28,29] by the Klingenberg group. All these studies were performed on rat liver mitochondria, which were the classical mitochondria for studying OXPHOS.

The main features to be briefly mentioned [22] are:(1)ADP is exchanged against intramitochondrial ATP with a one-to-one stoichiometry when mitochondria are actively respiring;(2)The only physiological substrates are ADP and ATP; surprisingly in their free forms, i.e., Mg-ADP and Mg-ATP, they are not recognized by the carrier;(3)Very high selectivity for the substrates: It does not transport AMP;(4)The highly charged and very hydrophilic nature of the large substrates;(5)The strong regulation by high electrical forces of the mitochondrial membrane potential of the highly charged substrates ADP^3−^ and ATP^4−^;(6)The kinetic parameters of the carrier are consistent with the mitochondrial ATP production and the cell nucleotide concentrations under physiological conditions;(7)The high levels of the underlying carrier in mitochondria to cope with the high demand for ATP in most aerobic cells;(8)The unique existence of highly specific inhibitors, as represented first by atractyloside (ATR) and then by other atractylogenin glucosides, in particular by carboxyatractyloside (CAT), greatly facilitating the characterization of the transport sites;(9)Another type inhibitor was represented by bongkrekic acid (BKA), which became a key element to elucidating the transport mechanism of transport of the ADP/ATP carrier at the molecular level;(10)Stabilized by these inhibitors toward detergents, in particular by CAT and thanks to its high levels in heart mitochondria, the ANT protein became the first carrier protein to be isolated in the native state;(11)The translocation mechanism of ANT was further elucidated and the involved conformation changes were demonstrated;(12)The first rational reconstitution of a carrier into vesicles was achieved, with the isolated ANT, and the transport parameters were determined in detail, in particular the differences between ADP and ATP and the control by the membrane potential and lastly;(13)The complete primary structure was determined by amino acid sequencing with large amounts of purified ANT protein.,. The elucidation of the complete primary structures of ANT in 1982 [28] and UCP1 in 1985 [29] by Klingenberg’s group, together with the sequence of the phosphate carrier, led to the initial definition of the homologous family of mitochondrial solute transporters.

## 3. ANT: Its Essential Features

### 3.1. A Brief History of ANT

The concept of a mitochondrial carrier for adenine nucleotides arose in the 1950s–1960s.

In a pioneering study from 1953, Siekevitz and Potter [30] established that two distinct pools of adenine nucleotides operate cooperatively in the OXPHOS pathway, one located in the mitochondrial matrix and the other in the cytosol. Soon after, Pressman [31] reported that “an extremely dynamic interchange of nucleotides” occurs between the two pools.

In 1962, Bruni et al. [32] and Vignais et al. [33] reported that ATR, a poisonous natural drug, specifically inhibits the OXPHOS of extramitochondrial, but not intramitochondrial, ADP, results that were also confirmed by several others groups [34]. The obvious conclusion was that ATR inhibits the entry of ADP into mitochondria, and consequently, the existence of a transporter for ADP was hypothesized.

In 1970, Henderson and Lardy [35] introduced BKA, a complex fatty acid derivative, as an ADP/ATP carrier inhibitor.

From that time to 1982, i.e., when the protein sequence of the ADP/ATP carrier was determined (see above), a large quantity of results dealing with the properties/characteristics of the carrier were collected.

### 3.2. ANT Exchanges Cytosolic ADP for Newly Synthesized Mitochondrial ATP

A paper by the mitochondrial carrier research group at the MRC MBU in Cambridge, UK, titled “The molecular mechanism of transport by the mitochondrial ADP/ATP carrier” and published two years ago by the journal *Cell* [36], represents a major step forward in understanding how the mitochondrial ADP/ATP carrier works at the molecular level. The carrier plays a vitally important role of transporting ADP into mitochondria and newly synthesized ATP out, replenishing the cell with energy. This process is vital to keep us alive, every second of our lives, for all of our lives, and makes us understand how mutations as well as effectors of various types can affect the function of this protein, resulting in a range of neuromuscular, metabolic, and developmental diseases.

Since we have only a small amount of ATP in our body, we need to remake it from the spent product ADP and phosphate using an enzyme complex, called ATP synthase, which is located in mitochondria. In this way, every molecule of ATP is recycled roughly 1300 times a day. For ADP to reach the enzyme, and for the product ATP to refuel the cell, each molecule has to cross an impermeable lipid membrane that surrounds the mitochondria.

The mitochondrial ADP/ATP carrier is involved in the transport of ADP into the mitochondria and of ATP out.

The ADP/ATP exchange follows Michaelis–Menten kinetics, with a Km of 1–10 μM for ADP and a Km of 1–150 μM for ATP for rat liver ANT [22]. The activity of this electrogenic transport is moderate, catalyzing the transport of 1500–2000 molecules per min. This relatively low specific activity is compensated by the high density of ANT, which represents about 10% of the total IM proteins.

### 3.3. Some Data about ANT Structure

ANT proteins have a characteristic structure with a tripartite repeat of ~100 amino acids and a total of six transmembrane domains that cluster in a generally cylindrical shape [37]. From a topological point of view, ANT proteins have carboxy- and amino-terminals that face the intramembrane space, with two exposed cytosolic loops (C1–C2), whereas the matrix side of ANT is composed of three hydrophilic loops (M1–M3) that are available for protein interaction and modification [37].

The concept of “induced transition fit” for carrier catalysis, by analogy with enzyme kinetics, was introduced by Klingenberg [38]. Briefly, he proposed a transition state between c- and m-states, according to which substrates, initially weakly binding to the c- or m-state, induce conformational changes that lead to tighter binding and the formation of the transition state. The carrier-substrate binding energy lowers the activation barrier for the formation of the transition state, then catalyzing transport.

ANT research has been highly informed by two unique classes of ANT inhibitors. The first is represented by ATR, a toxic molecule from the *Atractylis gummifera* thistle plant [39]. It, together with the chemically related CAT [37], locks the carrier in a cytoplasmic state (c-state) in which the substrate-binding site is accessible to the intermembrane space, which is confluent with the cytosol [40]. As such, ATR is only able to inhibit ANT function when applied to the exterior of mitochondria [22,41]. Notably, ATR inhibition can be overcome via competition with ADP, which drives ANT proteins back into the m-state [41], while CAT locks ANT-proteins in the c-state irreversibly.

The second class is represented by BKA, a respiratory toxin produced by the bacteria *Burkholderia gladioli* pathovar *cocovenenans* [42]. It is another potent inhibitor of ATP/ADP exchange, but, unlike ATR, BKA binds exclusively to the m-state conformation from inside the mitochondrial lumen [36,37,41], i.e., with the substrate-binding site accessible to the matrix side. BKA can be used to lock ANT proteins in the m-state [36,43].

The differential targeting of these two inhibitors has proved invaluable in determining the kinetic and structural features of ANT proteins.

#### 3.3.1. Cardiolipin Associates Strongly with the Mitochondrial ADP/ATP Carrier

Another important structural consideration of the ANT protein is that it tightly binds molecules of cardiolipin (CL), which is required for protein function and stability [44,45,46].

The importance of lipids in maintaining and establishing membrane barriers is accepted as dogma; however, at its most elemental, the lipid bilayer is much more than simply a random matrix providing structure for membrane proteins. In fact, within this pool of lipids, specific protein–lipid interactions critical for the structure, incorporation, and/or assembly of proteins occur [47].

As a signature phospholipid of mitochondria, CL acts as a determinant of membrane protein structure and function. It is almost exclusively found in association with the IMM, where it is synthesized, thus indicating that CL is critically important for this compartment, where it assumes a role of the utmost importance for the structural organization of the respiratory complexes in higher-order structures of functional importance [47,48]. CL also interacts with all of the major players in OXPHOS, including respiratory complexes I, III, IV, and V and the two members of the mitochondrial carrier family required for this process, ANT and the phosphate carrier [47]. In particular ANT excels in unusually strong interaction with CL [47] in a ratio of 3:1, as evidenced by 31P-NMR and ESR spectroscopic measurements [49], and by CL presence in crystal structures despite stringent washes during protein purification [50,51].

Although this aspect concerning the interaction between CL and ANT is beyond the scope of this review, because the presence of CL is critical for its functioning, deficits in ANT function should be critically evaluated in all the assorted pathologies that have been associated with disturbances in CL metabolism [52].

Consequently, it is intuitive to deduce that CL oxidation, reduced CL levels, and/or altered molecular composition can cause mitochondrial dysfunction associated with aging, ischemia and reperfusion, heart failure, and Barth syndrome [47]; deficits in CL synthesis and increased CL catabolism have been connected to diabetic cardiomyopathy [53]. Consistent with the concept that ANT may represent a downstream target subsequent to alterations to CL, CL peroxidation has been shown to inactivate mammalian ANT, resulting in apoptosis [54].

### 3.4. Does ANT Work as a Dimer or Monomer?

A key question that needed to be resolved is whether the mitochondrial ADP/ATP carrier requires dimerization to achieve its function. It is not surprising that these studies are often marred with controversy, in our opinion. In fact, all the experiments dealing with membrane proteins are technically challenging and many complicating factors often lead to incorrect interpretations. An aspect to take into account in these studies is that mitochondrial carriers are highly dynamic membrane proteins with very few polar interactions stabilizing their structures, causing major issues with almost every aspect of this analysis.

In one of their most recent works, Kunji and Ruprecht [55] dealt with this topic in a very accurate and detailed study.

As the authors remarked in their review article, all of the sequence, structural, biophysical, and functional data show that the mitochondrial ADP/ATP carrier exists and functions as a monomer. Studies on the structure clearly demonstrated that its structural fold is monomeric, lacking a conserved dimerization interface. Therefore, this is the only reference to which we refer the interested reader to issues concerning the ability of the protein to function as a monomer or as dimer.

### 3.5. ANT Isoforms

ANT is an abundant mitochondrial protein.

Humans possess four distinct ANT isoforms encoded by four genes (gene names in brackets): ANT1 (*SLC25A4*), ANT2 (*SLC25A5*), ANT3 (*SLC25A6*), and ANT4 (*SLC25A31*) [55], whose transcription depends on the cell type, developmental stage, cell proliferation, and hormone status. They have ~90% homology [56], except for ANT4, which has ~70% homology to the others [57]. The expression of the four isoforms is tissue specific and may depend on the proliferative capacity of the tissue and its energy requirements in terms of glycolysis or OXPHOS [58].

ANT1 is predominantly expressed in post-differentiated tissues such as heart and skeletal muscle [59]. ANT2 is specifically expressed either in undifferentiated cells, such as lymphocytes, or in tissues that are able to proliferate and regenerate, such as those of the kidney and liver [60]. The expression of ANT2, which has been shown to be growth-dependent, is considered a marker of cell proliferation [61]. ANT2 gene expression is downregulated in differentiated cell lines and remains unexpressed, or only slightly expressed, in most tissues [60]. ANT3 is ubiquitously expressed [60,62] and ANT4 is specifically expressed in lung and germ tissues [63,64].

In 1999, a systematic screen for proapoptotic genes led to the identification of ANT1 as a pro-apoptotic factor [65]. Apoptosis induced by ANT1 overexpression was accompanied by enhanced cytochrome *c* release and caspases-9 and -3 activation, indicating that its expression facilitates mitochondria-dependent apoptosis [65,66]. In support of its pro-apoptotic role in vivo, ANT1 transfection in breast adenocarcinoma cells in a nude mouse model induced apoptosis, increased *Bax* expression, and finally stimulated tumor regression, validating the hypothesis that ANT1 could be a potent therapeutic target for the treatment of cancer [56]. The direct implication of ANT in apoptosis offers one more reason to focus on ANT in cancer development. Furthermore, ANT works with Bax, cyclophilin D (CyP-D) of the matrix, and VDAC of the OMM to form a permeability transition pore (PTP) (see below, Section 3.6.2), a lethal pore during apoptosis [60].

Since the expression of ANT isoforms varies with mitochondrial energy metabolism, overexpression of ANT isoforms reveals variations in apoptotic behavior. Therefore, the overexpression of ANT1 and ANT3 can increase ATP export from the matrix and lead to mitochondrial de-energization that induces apoptosis. Conversely, in tumor cell lines, the import of ATP and the so-called matrix conformation of ANT are favored; therefore, the ANT2 isoform can be a direct inhibitor of the PTP function of ANT [67] and then considered an anti-apoptotic oncoprotein [68,69] that functions as an inhibitor of mitochondrial membrane permeability.

In differentiated adult cells, ANT1 is highly expressed and ANT2 expression is minimal [69], whereas when it turns cancerous—switching from aerobic respiration to aerobic glycolysis—it downregulates *ANT1* and upregulates *ANT2* expression, which is linked to the rate of glycolytic metabolism and is an important indicator of carcinogenesis. Under OXPHOS impairment conditions, ANT2 imports glycolytically produced ATP into the mitochondria. In the mitochondrial matrix, the F1Fo-ATPase complex hydrolyzes the ATP, pumping out a proton into the intermembrane space. The reverse operations of ANT2 and F1Fo-ATPase under glycolytic conditions contribute to maintaining the mitochondrial membrane potential, ensuring cell survival and proliferation. Indeed, in the majority of cancer cell lines, ANT2 is very high [69], unlike that of ANT1. ANT3 expression is low across the board.

Unlike the ANT1 and ANT3 isoforms, ANT2 is not pro-apoptotic.

If ANT1 and ANT3 are associated with aerobic respiration, thus exporting the ATP produced by OXPHOS from the mitochondria into the cytosol while importing ADP [60], by contrast ANT2 is associated with aerobic glycolysis [60]. Its expression is a marker for rapid proliferation and/or cancer. Hence, specific inhibition of ANT2 is a prospective anti-cancer strategy.

That cancer cells use mitochondrial ATP synthase to hydrolyze glycolytic ATP in order to maintain ΔΨ is certain; however, whether in cancer cells, the entry of glycolytic ATP into mitochondria occurs by an ANT-dependent pathway or something other than ANT [70] is unknown to date. Thinking about it, it could be through the electroneutral ATP-Mg/Pi carrier, which exchanges ATP (Mg-ATP^−^) for Pi (HPO_4_^2−^) [57], but this is not the only hypothesis. In fact, since the human genome encodes 48 different mitochondrial carriers [71] and a big proportion of them have not yet had their function assigned, it could be that one or more of these “orphan” transports facilitates ATP entry into mitochondria.

In a recent paper, Forrest [57] identified molecular targets and operations unique to cancer cells; in particular, the author, suggesting that ANT2 does import ATP into mitochondria of cancer cells, predicted that ANT2 is inversely orientated in the IMM compared to normal cells, making it resistant to inhibition by CAT and BKA, which are both inhibitors of ANT. This provides the explanation for the inability of these drugs to block ATP entry into the mitochondria of cancer cells [70]. It follows that ANT2 has a different drug interaction profile, which can be leveraged for a selective cancer therapy. A further distinction is that this opposite orientation of ANT2 may make it less predisposed to joining in to make the permeability transition pore (PTP) in apoptotic scenarios [57].

### 3.6. ANT: A Multitasking Protein

In addition to functioning as exchanger, a large body of experimental evidence demonstrates that ANTs are multifunctional proteins that are poised to participate in several aspects of mitochondrial biology.

We believe that this aspect deserves at least to be mentioned, albeit briefly. Therefore, here we describe how ANTs participate in the formation of mRC supercomplexes, as well as how they constitute a pore-forming component of the mitochondrial permeability transition pore (mPTP), a structure that forms in the IMM that is thought to underlie regulated cell death. ANT proteins are also critical for promoting mitophagy.

#### 3.6.1. ANT Is Part of mRC Supercomplexes

There is solid evidence that mitochondria coordinate the activity of diverse enzymatic processes involved in mitochondrial oxidative metabolism by grouping proteins into supercomplexes to locally increase efficiency towards the goal of producing ATP.

“ATP synthasome” is one mitochondrial supercomplex. It contains the F1Fo-ATP synthase as well as ANT and phosphate carrier proteins in a 1:1:1 stoichiometric ratio so that the two transporters that have to do with OXPHOS are proximal to the site of ATP production [37,72]. It is thought that this structure ensures a robust supply of inorganic phosphate and ADP to the F1Fo-ATP synthase as well as rapid export of produced ATP.

The respirasome, the main “respiratory chain supercomplex” (mRCS), involving electron chain complexes I, III, and IV [37,73], also associates with ANT.

Although ANT association with RCS is not required for the individual function of either electron chain complex or ANT proteins themselves, it has been noted that when RCS assembly is inhibited, overall ATP/ADP exchange rates are decreased [73], suggesting that RCS integrity is important to ANT function. Interestingly, many if not all of these interactions depend on CL (see above, Section 3.3.1). In particular, two years ago, Parodi-Rullán et al. [74] sought to elucidate whether ANT knockdown affects RCS formation in H9c2 cardiomyoblasts. Their study confirmed that ANT is involved in RCS assembly, although RCS may not be solely dependent on ANT. Moreover, ANT can physically interact with RC complexes I, III, and IV [73] and thus be involved in the respirasome structure or play a regulatory role in the formation/maintenance of mRCS assembly. However, the same authors declared that further studies are required to elucidate the role of ANT in the structural integrity and regulation of mRCS and other mitochondrial supercomplexes (e.g., ATP synthasome) in cardiac cells.

#### 3.6.2. ANT Is a Component of mPTP

The reversible Ca^2+^-dependent increase of IMM permeability to solutes with molecular masses up to about 1500 Da is defined as a mitochondrial permeability transition (mPT) and is mediated by the mitochondrial permeability transition pore (mPTP) [75]. The proposal that the permeability transition is linked to the reversible opening of a pore rather than to unspecific membrane damage is based on the inhibitory effect of cyclosporin A (CsA) used in nanomolar concentrations [76]. mPTP opening causes both (i) mitochondrial swelling and the rupture of the outer membrane, a process leading to the release of soluble pro-apoptotic proteins from the intermembrane space and (ii) the dissipation of the mitochondrial H^+^ gradient, thus collapsing the Δψ and in turn abolishing the mitochondrial capacity to produce ATP [37], a real energetic catastrophe that invariably leads to cell death.

It is commonly believed that mPTP is a multi-protein system. Originally, only regulatory components were identified. The first unambiguously established component was CypD (in the matrix), which remains the only protein whose involvement in mPTP pore formation and activity regulation is undisputed [77]. The identity of other proteins comprising the PTP is to date debated; however, mPTP is usually described as a polyprotein complex located at the mitochondrial contact site formed from the voltage-gated anion channel (VDAC, in the OMM), members of the Bax/Bcl2 pro and antiapoptotic protein family [78], and additional proteins such as HK (in the cytosol), creatine kinase, peripheral benzodiazepine receptor (PBR, in the OMM) [79], and the phosphate carrier [80]. To these must also be added ANT, widely accepted as a candidate for PTP formation and first proposed by Halestrap and Davidson in 1990 [81].

Therefore, ANT appears to act as a bi-functional protein, contributing, on the one hand, to a crucial step in aerobic energy metabolism, the ADP/ATP translocation, and on the other hand, converting into a pro-apoptotic pore under the control of onco- and anti-oncoproteins from the Bax/Bcl-2 family. In particular, in response to multiple pro-apoptotic stimuli, Bax moves from the cytoplasm to mitochondrial membranes. Upon insertion into OMM, Bax can form a pore, presumably via oligomerization with itself or with its homologs (Bak, Bad, t-Bid) and/or interactions with mPTP components, such as VDAC and/or ANT [82]. As reported by Belzacq et al. [82], in a hypothetical cooperation model Bcl-2 would stimulate the translocator function of ANT, whereas Bax, after a conformational change induced by ATR, would stimulate the pore function of ANT. On the contrary, Bax would diminish the translocator function and ATP, ADP, and Bcl-2 would inhibit the pore function of ANT. Shortly after, the same authors, utilizing a variety of different systems (proteoliposomes containing purified ANT, Bcl-2, and/or Bax; isolated mitochondria and mitoplasts; intact cells), reported that Bcl-2 and Bax, in addition to modulating pore formation by ANT, importantly also influence the enzymatic function of ANT as an ADP/ATP antiporter, thus shedding new light on the apoptosis-regulatory functions of Bcl-2/Bax-like proteins [67]. These observations are corroborated by the fact that several ANT ligands and inhibitors modulate mPTP opening activity [37].

However, although a number of studies have been carried out with the aim of demonstrating unequivocally the involvement of the ADP/ATP carrier in mPTP formation, the conclusions are controversial.

In this regard, we refer to the latest review on this topic by Professor Bernardi [78,83,84], an example of excellence in the field, who has taken care over the years to follow the discoveries regarding the mPTP since the idea appeared in 1990 [81].

The hypothesis of ANT as a component of the mPTP was first questioned by Kokoszka et al. [85] because an mPT was detected after ablation of the genes encoding for ANT1 and ANT2 isoforms, expressed in mouse liver mitochondria, but recently Karch et al. [86] noted that the same mice overexpress the ANT4 isoform, which might have provided a permeabilization pathway. Readdressing the role of ANT in the mPT was obtained by results from experiments performed with *ANT1/ANT2/AN*T4 triple-knockout mice mitochondria: a PTP still existed, suggesting that the PTP forms in the absence of ANT.

However, further genetic ablation of the *Ppif* gene (which generates CyP-D null mitochondria) or treatment with CsA (which inhibits CyP-D) finally prevented any Ca^2+^-dependent permeabilization, suggesting that mitochondria possess at least two pathways for Ca^2+^-dependent permeabilization and that they are both inhibited by CsA [86]: one in the absence of ANT, which is not inhibited by BKA [83,87], and one in the presence of ANT, which is inhibited by BKA [87].

It seems that despite its role as the main energy-producing enzyme, it is precisely the ATP synthase that plays a role in the IMM depolarization caused by the channel activity of the mPTP. However, this aspect of the question concerning PTP, although of considerable importance, is a problem beyond our competence and therefore we refer the interested reader to specific readings [77,78,83,84,88].

#### 3.6.3. ANT Promotes Mitophagy

Mitophagy is a particular form of autophagy, which is a quality control process [89] that cells have designed to sequester and remove damaged mitochondria and thus to defend against dysfunctional mitochondrial activities.

In fact, as the result of cellular senescence or pathological processes, mitochondria can damage cells both by failing to perform their critical functions (e.g., ATP generation, maintaining normal cell signaling) and by actively stimulating hazardous processes (e.g., release of toxic ROS, apoptosis, increasing cellular calcium levels) [90].

Briefly, mitophagy is a process triggered by a loss of mitochondrial membrane potential that then initiates the accumulation of two proteins, phosphatase and tensin homolog-induced kinase 1 (PINK1), and the E3 ubiquitin ligase Parkin on the outer surface of the damaged mitochondrion. In response to this event, the mitochondrion is wrapped in a membrane that isolates it from the rest of the cell, transforming it into an autophagosome; subsequently, the latter fuses with a lysosome, degrading the enclosed organelle [91]. It follows from this that mitophagy, far from being a deleterious process that destroys mitochondria, is a beneficial process; therefore, mutations that alter the function of the PINK1/Parkin proteins sometimes result in severe disease [90].

In healthy mitochondria, PINK1, a key component of the mitophagic machinery that marks dysfunctional mitochondria for degradation [92], is translocated via the TOM and TIM23 complex into the mitochondrial matrix and degraded. Since PINK1 translocation into the mitochondrial matrix via TIM23 requires Δψ, it follows that during mitochondrial dysfunction, which causes loss of Δψ, PINK1 cannot enter the matrix and then accumulates on the OMM, where it recruits the E3 ubiquitin ligase Parkin, thus initiating the mitophagic process [92].

Recently, Hoshino et al. carried out a sophisticated and meticulous work, published in *Nature* in 2019 [93], in which, using multiple mitophagy reporter systems and pro-mitophagy triggers, they identified numerous components of Parkin-dependent mitophagy, including, surprisingly, ANT. At the molecular level, ANT is required for inhibition of the presequence translocase TIM23, which leads to the stabilization of PINK1, in response to bioenergetics collapse.

It seems that ANT is required for mitophagy in several cell types.

If pharmacological inhibition of ADP/ATP exchange promotes mitophagy independently of its transport activity, genetic ablation of ANT paradoxically suppresses mitophagy in response to uncoupling or treatment with metabolic toxins, with consequent profound accumulation of aberrant mitochondria.

Ideally, ANT is a sophisticated sensor of bioenergetics failure from either proton gradient collapse or failure to carry out OXPHOS. It suppresses TIM23 translocation of PINK1, thus facilitating the stabilization of PINK1 on the OMM and activating mitophagy. The fact that PINK1 translocation continues to occur in uncoupled mitochondria lacking ANT proteins represents a substantial challenge not only to the understanding of mitophagy, but also to the current model of how proteins are transported into the mitochondrial matrix [94].

In conclusion, the findings show that the protein has two separate functions: ATP generation and quality control. In fact, although it is known that the mutations in the *ANT* gene cause diseases, they do not impact ANT’s ability to produce chemical energy, but rather cause defective quality control, an important piece of knowledge to start thinking about therapeutic approaches that may improve quality control.

As such, ANT proteins are likely the key integration sites for mitochondrial health, function, and cell death. Moving forward beyond 2020, further studies will be needed to define the regulatory mechanisms of these processes and to better characterize the different modes of action of ANT.

### 3.7. ANT Kinetic Determination by Using the ATP Detecting System

ANT kinetics have great physiological relevance since it is likely that the kinetic parameters of the carrier are a major determinant of the rate of cytosolic ATP utilization in situ. Most kinetic studies—in the late 1980s—employed isotopically labelled substrates of the carrier and measured the uptake/release of isotopes into/from the mitochondrial matrix. However, many problems were encountered when attempting to interpret data.

It soon became clear to everyone that direct accurate measurements of initial rate are almost impossible to achieve because the transport is rapid, the exchangeable pools are small, and the inhibitor stop method technique inhibits at variable rates dependent on conditions of energization and substrate concentration.

In order to circumvent the problems inherent in measuring ANT kinetics in intact mitochondria, Krämer and Klingenberg [13,95,96,97,98] measured ANT kinetics in a reconstituted system composed of artificial liposomes and the isolated carrier. Unfortunately, even in this case, various problems related to time-dependent variations, to K^+^ gradient used to assess, and so on showed up.

Several other methods beyond the isotopic one (method 1) for determining the ATP–ADP steady-state exchange rate mediated by ANT have been described in the past.

Some of these directly measure ADP and/or ATP by (2) thin-layer chromatography [99] or (3) high-performance liquid chromatography [100]. Another employ coupled reactions that yield an end-product that can be detected (4) luminometrically, detecting chemiluminescence upon ATP-dependent oxidation of luciferin, catalyzed by firefly luciferase [101]. Furthermore, there are techniques that employ (5) fluorescent derivatives of nucleotides in order to estimate the rate of release of the fluorescent molecule upon ANT-mediated exchange for ADP [102].

Among the aforementioned methodologies, the radioactive method, a very sensitive assay that is already operational with small amounts of mitochondrial protein, requires the handling of radioactive material, and methods 2 and 3 require expertise in using specialized equipment. Above all, they are essentially end-point assays.

Only methods 4 and perhaps 5 could be considered on-line measurements. However, for its part, method 5 relies on materials that are not commercially available and are therefore not appealing to the broader scientific community. Method 4 (the luciferase method), if used as an on-line method, suffers from several drawbacks: First, there is a lack of constant proportionality between ATP concentration and luminescence caused by product inhibition of the luciferase reaction by oxyluciferin; second, for the reliable estimation of the kinetics of ATP formation, the endogenous ATP concentration prior to the assay needs to be measured; third, the required use of a low ionic strength medium and room temperature restricts experiments to conditions that are nowhere near physiological. Although some of the disadvantages of this method have been overcome by various modifications [103], it is certainly not a suitable method for the purpose.

Overall, for a comprehensive appraisal of methods to measure ATP–ADP exchange rates in mitochondria and reconstituted systems, the reader is referred to the review by Martin Klingenberg [22].

It was 1988.

Taking a cue from measurements and kinetic determinations of other transporters [16] for which a variety of metabolite detecting systems (consisting of enzyme/s and cofactor/s designed to selectively ascertain outside mitochondria the appearance of molecules derived from the mitochondrial metabolism of the taken-up substrate) have been developed, the idea of applying the same criterion to ANT tickled us a lot and proved to be successful.

Obviously, this carrier plays a primary role in cellular metabolism because the function of the mitochondrion is—by definition—to supply the cell with ATP, which is also the universal energy signal of metabolism, being generated or consumed in every metabolic pathway. Therefore, the carrier is important for the regulation of cellular metabolism and hence—as we will see—it is involved in various pathological states (see below, Section 4).

Of course, in this case, unlike the other mitochondrial carriers, we were faced with a “complex” case: if a correct study of the permeability properties of the mitochondria could not disregard their integrity and functionality, making it essential to verify the mitochondrial coupling before each experiment, in this case the question was scorching. It was possible to follow the ADP/ATP antiport exchange by using exclusively mitochondria in which OXPHOS occurred. Therefore, we were in conditions where the proton gradient would be consumed both to operate the carrier and for the OXPHOS.

The appearance of ATP in the extramitochondrial phase, revealed by a specific detecting system, is the result of a process consisting of several stages: (1) uptake of ADP via ANT in exchange with endogenous ATP; (2) synthesis of ATP via ATP synthase, which makes use of the energy resulting from the oxidation of endogenous substrates; and (3) export of the newly synthesized ATP in exchange with ADP.

Using rat liver mitochondria (RLM), a method was developed to continuously monitor the ATP efflux from mitochondria, which occurs by adding ADP to the mitochondrial suspension incubated in the presence of glucose, NADP^+^, hexokinase (HK), and glucose-6-phosphate dehydrogenase (G6PDH) by revealing the appearance of ATP in the extramitochondrial phase photometrically as an increase of absorbance at 340 nm due to NADP^+^ reduction [104], taking into account that there is a stoichiometric ratio of 1:1 between effluxed ATP and formed NADPH (Figure 1).

Really, the experimental strategy is complex. In fact, ATP production by mitochondria incubated with ADP can occur both via ATP synthase activity in OXPHOS and via adenylate kinase (ADK), which is localized in the intermembrane space. Thus, in order to separate the contribution of these two pathways to ATP production and appearance outside mitochondria, we used CAT, oligomycin (OLIGO), and Ap5A to inhibit ANT, ATP synthase, and ADK, respectively.

We observed that in the presence of Ap5A, the rate of ATP appearance was completely inhibited by adding either CAT or OLIGO. Instead, when the same experiment was carried out photometrically in the presence of CAT (5 μM) and OLIGO (2.5 μM), thus monitoring only the ADK-dependent ATP production, ATP was formed as a result of ADP addition. As expected, the addition of Ap5A (20 μM) completely blocked the rate of NADPH formation.

The rate of NADP^+^ reduction showed a hyperbolic dependence on ADP concentration and proved to be a measure of the activity of the ADP/ATP translocator in the presence of Ap5A, used to inhibit the adenylate kinase [104,105], as established by the determination of limiting step of the global process (see below).

Importantly, for the first time, the ADP/ATP exchange was studied by using respiring and phosphorylating mitochondria, i.e., under roughly “physiological” conditions, since the intramitochondrial ATP concentration is not enhanced by a loading procedure, but is derived only from the uptake and phosphorylation of externally added ADP.

Furthermore, assuming that the available energy is not limiting, the flow rate of ATP will depend on either the rate of the antiporter or that of the ATP synthase.

Thus, to find out whether ANT or ATP synthase determines the rate of ATP appearance outside mitochondria, this activity was measured in the presence of CAT and OLIGO. The former inhibits the transport but not the synthesis of ATP; on the contrary, the second inhibits synthesis without inhibiting the carrier.

By measuring the rate of the ATP efflux as a function of the concentration of each of the two inhibitors, the inhibition was immediate and increasing for the inhibitor of the reaction representing the limiting stage, i.e., CAT.

Although OLIGO inhibits ATP synthase, it is unable to inhibit the global process of ATP efflux until the rate of ATP synthesis via ATP synthase inhibited by OLIGO becomes lower than that of the carrier. Plotting the reciprocal of the speed measured as a function of the concentration of the non-penetrating inhibitor, the intercept of the straight line at the ordinate axis is a measure of the transport activity in the absence of an inhibitor, indicating that the ATP/ADP exchange determines the rate of ATP efflux. On the contrary, with OLIGO, not all points are at the same line: the *y*-axis intercept (equal to zero (OLIGO)) of the extrapolated lines, differed (1/V lower) from the activity measured at zero (OLIGO). Thus, we were able to conclude that the rate of ATP appearance outside mitochondria is governed by the rate of ADP/ATP exchange via ANT rather than the rate of ATP synthesis via ATP synthase.

This is one case in which (see above), as expected, the determination of the limiting stage could not be done through the use of a detergent since OXPHOS requires intact mitochondrial membranes. That is why, since ANT activity depends on the mitochondrial membrane potential mΔΨ, measurement of its activity was always preceded by that of the mΔΨ. The choice of substrates, the quality of mitochondria, and the buffer composition, to name a few, all affect mΔΨ, and as extension of this, the ADP–ATP exchange rate. It follows that another aspect to take into account when applying this method is the fact that, under the adopted experimental conditions, the energy for ATP synthesis was given by the oxidation of endogenous substrates. Sometimes, for experimental needs, to ascertain that mitochondria themselves can drive ATP synthesis without any limitation due to energy deficit, we energized mitochondria present in cell homogenates by adding β-hydroxybutyrate (5 mM), a respiratory substrate that enters mitochondria via diffusion with an increase in membrane potential. If the rate of ATP production was not significantly increased under these conditions, we concluded that the mitochondria themselves could generate the electrochemical proton gradient needed for ATP synthesis. Conversely, prevention of ΔμH^+^ formation from endogenous substrates by the addition of a cocktail of the electron flow inhibitors rotenone, antimycin A, mixothiazole, and cyanide resulted in blocking ATP production.

Furthermore, extreme care was taken to use enough HK/G6PDH-coupled enzymes to ensure a non-limiting ADP-regenerating system for the measurement of ATP production. Therefore, as controls, it was always verified that (i) the coupled enzymatic system used to detect ATP outside mitochondria was not rate limiting per se by showing that the addition of further ATP resulted in an increase in the rate of absorbance, (ii) whereas no rate increase occurred following the addition of both substrate and enzyme components of the ATP D.S.

## 4. ANT: In Health and in Disease

Before going into the specific topic whose pivot is the impairment of ANT in different pathological conditions, we want to trace in a few lines the path carried out by this protein in the last 30 years.

Surely an important milestone was achieved in 1996, when it was discovered that ANT, along with other proteins, also functions as a non-specific pore upon stimulation by ions and a variety of molecules, such as calcium (Ca^2+^) and ROS, as shown by the reconstitution of native purified ANT into liposomes [106]. A second important breakthrough highlighted the critical involvement of ANT as a major player in the execution of the mitochondrial apoptosis pathway in various pathophysiological and experimental models [56,107], according to which ANT defined cell fate by cooperating with pro-apoptotic proteins such as Bax, viral proteins (for example, viral protein R (Vpr) from HIV-1, PB1-F2 from influenza), or by participating in the mitochondrial polyprotein complex in diverse model systems such as human carcinoma cell lines, cardiomyocytes, and neurons [56,108]. More recently, the fact that each ANT isoform has a specific role in influencing cell fate, notably in cancer cells [56,109], represented another flag raised by the research on ANT.

From this roundup of milestones it can be deduced that ANT is a bi-functional protein, and that it harbors two opposite functions, both of which are intricately involved in the control and regulation of cell fate [56]: a vital function important for the maintenance of mitochondrial function to ensure the life of the cell, and another lethal one, leading the cell to death.

It follows that it is intuitive to think that impairing its function leads to severe diseases.

At this point in our study, beyond briefly reviewing human diseases associated with mutations or altered expression of ANT, we will focus on some disease states in which ANT function is compromised. In addition, we will describe some particular conditions—either pathological or conditions in which the effect on ANT by an effector with which mitochondria or cells are treated is observed—in which ANT was in the spotlight, thus leading to monitoring its dysfunction by applying the method of the ATP detecting system (ATP D.S.).

### 4.1. Mutations or Altered Gene Expression Associate ANT with Human Diseases

ANT1, one of the four isoforms (see Section 3.5), has been so far the only to directly cause mitochondrial diseases.

Based on investigations of mitochondrial dysfunction it is evident that aberrant ANT1 activity causes human disease, certainly as a result of diminished activity—as in autosomal dominant progressive external ophthalmoplegia (adPEO) and Senger’s syndrome—and possibly also due to its overexpression in facioscapulohumeral muscular dystrophy (FSHD) [110].

However, the discovery that ANT1 dysfunction can cause adPEO not only revealed a surprising role in mtDNA stabilization, but also provided one example of an mtDNA deletion disorder caused by an autosomal gene defect.

In Senger’s syndrome, an autosomal recessive disease characterized by hypertrophic cardiomyopathy, mitochondrial myopathy, lactic acidosis, and congenital cataracts [111], it appears likely that ANT1 depletion is secondary to a mutation in a gene that functions in processing the translocase. No mutations have been found in ANT1, causing speculation that the transcription, translation, or posttranslational modification of ANT1 may be affected [112]. In fact, this is so: the loss of AGK, the mitochondrial acylglycerol kinase involved in lipid metabolism, may result in a decrease in ANT by affecting its biogenesis.

In contrast to ANT1 deficiency, overexpressed ANT1 may contribute to the pathogenesis of other diseases such as FSHD, characterized by the cumulative progression of muscle weakness in the face, feet, shoulders, and hips, along with occasionally sensorineural hearing loss [113]. FSHD is a highly variable autosomal dominant neuromuscular disorder. Although overexpression of *FRG1*, a gene involved in pre-mRNA splicing, was proposed as being responsible for FSHD, other studies proposed the overexpression of ANT1 and increased oxidative stress in FSHD muscles [110].

At this point, we would like to briefly describe one of the latest studies performed by Kliment et al. [114], just to underline how research on ANT is still fervent 30 years after its discovery and also to introduce the ANT2 isoform, which is involved in the development of chronic obstructive pulmonary disease (COPD). The authors observed that ANT2 expression is reduced in lung tissue from patients affected by COPD, and in a mouse smoking model. ANT1 and ANT2 overexpression resulted in enhanced oxidative respiration and ATP flux, stimulating airway surface liquid hydration by ATP and maintaining ciliary beating after cigarette smoke exposure, which are key functions of the airway. Therefore, the canonical protein, i.e., ANT (with ANT1, 2, and 3 present in human lungs), not only regulates mitochondrial metabolism but also plays a central function in airway epithelial biology, notably airway hydration, which promotes ciliary function.

### 4.2. ANT: Its Metabolic Role in Various Diseases

Unlike some more specialized reviews on the “ADP/ATP carrier” topic, the purpose of this review section is to focus only on the analyses of the functional activity of ANT, allowing us to assess its role in most varied pathophysiological contexts. Therefore, we selectively report here only those studies in which mitochondria, isolated from tissues/cells or present in the homogenates, are respiring and phosphorylating, where ANT works in own environment, the only condition that allows us to assess its role in any pathophysiological context that mimics the real situation in which the cell finds itself.

We are of the opinion that two general approaches can be used to investigate mitochondrial bioenergetics. The first approach requires the isolation of mitochondria from cells or tissues of different origins to design experimental conditions and protocols aimed at ensuring their integrity and mirroring the functional activity—as accurately as possible—of the environment of the intact cell. The great temptation is to modify the experimental conditions in order to amplify the phenomenon to be studied.

Instead, the second approach is to work with intact cells, where the gain of physiological relevance is balanced by the greater complexity of the preparation and the difficulty of accessing the mitochondria in situ. As we will see later, the research in our group focused on the development of techniques to improve access to the bioenergetics of mitochondria in situ by miniaturizing certain experimental approaches that are normally used to carry out measurements on isolated mitochondria.

With the intention of making a real reconnaissance of studies in which ANT appears, we will start with 1988, the historic year in which the ATP detecting system was developed and tested, and—through various types of studies regarding the functioning of ANT in various conditions ranging from apoptosis to necrosis (as regards neurons) to the condition in which there is an explosion of oxidative stress, or even the condition in which laser light or a photosensitization condition with specific drugs manifest their effect on the translocator, and again to ANT as a target of the two Alzheimer’s proteins and so on—we will get to the present day (Figure 2).

We will look at them one by one. For each of them—we would like to reiterate—we resorted to the procedure that allows for the continuous monitoring of ATP efflux from mitochondria incubated with ADP, using the ATP detecting system.

#### 4.2.1. ANT as a Target of Light

As mentioned earlier, the reason why in 1988 we wanted to develop a measurement of the ADP/ATP carrier was the evidence accumulated in previous years that mitochondria were targets of light irradiation (for ref. see [115,116,117]).

Passarella et al. [116,117] proved that irradiation of isolated RLM with a helium neon laser causes both an increase in the electrochemical proton gradient and ATP extra-synthesis. Given that cytosolic ATP availability could be dependent on the rate of ADP/ATP exchange, investigation into the possible effects of He-Ne laser irradiation on the activity of the ADP/ATP translocator seemed worthwhile. Therefore, in this first study, the development of such a method was presented as well as its first application to investigate the ADP/ATP antiport in mitochondria irradiated by a He-Ne laser. Irradiation of isolated RLM was also found to increase the rate of ADP/ATP exchange [104], likely due to the increase in the electrochemical proton gradient that occurs owing to the irradiation of mitochondria.

Some years later, after a large body of work had been accumulated showing that He-Ne lasers stimulate both biogenesis and bioenergetics of mitochondria, specifically determining an increase in mitochondrial ATP synthesis, we made a correct observation: endogenous adenine nucleotides, substrates involved in ATP synthesis, are also themselves irradiated under conditions in which ATP extra-synthesis occurred, that is, when the mitochondrion was irradiated, as in [104]. Therefore, the question that arose was intriguing: could a He-Ne laser light affect the biochemical features of AMP, ADP, and ATP? It had already been observed that NADH was sensitive to ruby and He-Ne laser irradiation and that irradiation of substrates and enzymes with non-coherent light proved to change their biochemistry [118]. Therefore, adenine nucleotides irradiated with 3 Joules/cm^2^ fluence (10 mW/cm^2^ fluence rate) showed altered biochemical behaviors when used as substrates for certain mitochondrial reactions in isolated RLM. It was observed that only certain enzymes are able to recognize irradiated adenine nucleotides, and that this property is not a substrate property, but rather concerns the substrate–enzyme complex [118]. We arrived at this conclusion since different effects were observed for the same irradiated substrate in different reactions. As a result of irradiation, ADP affinity for the ADP/ATP carrier increased, whereas it decreased when the ADK reaction was studied. No difference between irradiated or non-irradiated ADP was found in the ATP synthase reaction.

In addition, mitochondria were the target of photosensitization, a reaction to light mediated by a light-absorbing molecule that causes biochemical changes, ultimately leading to cell death [115,119]. Photodynamic therapy, which precisely consists of treatment with a hematoporphyrin derivative followed by photoradiation with visible light, was a promising approach to the treatment of various cancers. The first step in the photoinduced process is light absorption; then the photosensitizer, excited by light, is converted to a triplet excited state; this latter species leads to the formation of singlet oxygen and radicals that oxidize biomolecules. If, on the one hand, hematoporphyrin derivatives were, at that time, the most commonly used drugs for fluorescent visualization and photosensitization of various tumors [120,121], and in particular photofrin II (PF II) was indicated as the most active component by [120], on the other hand, mitochondria were indicated as the primary target of PF II photodamage [121]. In support of this, a depletion of the intracellular ATP pool was found in cancer cells in tissue culture or in situ, possibly due to the impaired mitochondrial function owing to PF II-induced photosensitization, which likely produces ROS, mainly singlet oxygen, that damages cellular components, leading to cytotoxicity [122].

It was a duty and of undoubted interest to clarify the mechanism of interaction between mitochondria and hematoporphyrin. Therefore, as part of the Researchers’ Exchange Agreement between the National Research Council (CNR, Italy) and the Institut National de la Santè e de la Recherche Medicale (INSERM, France), this topic—project research title: “The effect of hematoporphyrin photosensitization on mitochondrial and cellular activities”—was the object of study by A.A., in quality by Chercheur Associè, during her stay in Paris. Although the results obtained proved to be extremely interesting, now we leave them aside to focus on the question of photosensitization action on ANT. Strong non-competitive inhibition of the ADP/ATP exchange carrier was observed in isolated RLM when challenged with hematoporphyrin-sensitized photodynamic action, showing that the adenine nucleotide carrier is a major target of photodynamic action, which causes OXPHOS impairment [123]. In order to understand why PF II photosensitization was able to cause impairment of ANT activity, in a dedicated study, Atlante et al. [124] observed that photodynamic action exerted by PF II on ANT, a carrier protein sensitive to mercurial compounds able to bind to mitochondrial thiols—the target of singlet oxygen [125]—was prevented if mitochondria were treated with mersalyl, thus suggesting that SH groups present in carrier molecules are the functional groups affected by PF II photosensitization.

#### 4.2.2. ANT as a Target of Molecules

Within the therapy of some pathologies involving mitochondria, structurally different pharmacological agents, including antitumors, immunosuppressants, anesthetics, etc, have been designed to specifically affect mitochondrial functions. In other cases, the same drugs (such as anti-viral drugs) with primary targets in other cellular locations were found to impair mitochondrial functions as side effects.

Really interesting is the study conducted on 3-azido-3′-deoxythymidine (AZT), the most widely used antiretroviral drug in acquired immunodeficiency syndrome (AIDS) therapy, by D.V. during her PhD [126,127]. It seemed interesting to us to propose it in this context because this study is an interesting example in which, although the AZT molecule competitively inhibits ANT, it is not taken up into the mitochondria through it. ATP D.S. as a method to follow ADP/ATP exchange is well suited in this situation to better highlight ADP/ATP translocator impairment by AZT as one of the biochemical processes responsible for ATP deficiency syndrome, the main severe side effect associated with antiviral drug therapy.

First, the authors showed that AZT per se can enter mitochondria, as shown by both isotopic and HPLC measurements, and, more, that its transport occurs in a carrier-mediated manner. Furthermore, although externally added AZT was found to inhibit the ANT-mediated ADP/ATP exchange in both mitochondria and mitoplasts, the drug does not use the ADP/ATP translocator to enter the organelles. This conclusion is based both on the failure of AZT to cause an efflux of intramitochondrial ATP, and more importantly, on the insensitivity of AZT uptake to CAT, a powerful inhibitor of the ADP/ATP carrier.

The impairment of the ADP/ATP translocator could well be the biochemical process that exerts a direct effect on OXPHOS, which in turn could be responsible for the ATP deficiency syndrome induced in cells treated with AZT and which is proposed to be a severe AZT side effect in AIDS therapy [128,129]. If AZT uptake in mitochondria and AZT inhibition of ADP/ATP transport play a significant role in ATP deficiency syndrome, then the discovery of a proper inhibitor of AZT transport in mitochondria, as well as a proper AZT chemical modification that could prevent ADP/ATP carrier impairment and/or its uptake into mitochondria, could be valuable goals for further research.

#### 4.2.3. The Rate of ATP Export Declines with the Onset of Hypertension and in Aging

From the liver we move to the heart, where maintaining a high ATP supply is critically important for maintaining cardiac performance since the amount of ATP made and used per minute is many times greater than the size of the ATP pool. Furthermore, since ATP availability in cardiac cells is dependent mainly on OXPHOS, which can occur only in intact mitochondria, we investigated the ATP production made in coupled and respiring mitochondria with the aim of ascertaining the process controlling cell ATP availability. Furthermore, there was reason to suspect that cell ATP supply could be modified in the heart of both normotensive and spontaneous hypertensive aging rats. It was prompted us to investigate how ATP availability outside mitochondria is regulated in mitochondria from rat heart left ventricles (MLV) in hypertrophy and hypertrophy/hypertension states and whether and how it changes in aging.

In this as well as in similar cases, to understand the relationships between the different cellular components in a system biology framework, it may be more appropriate to consider functional products rather than genes, in light of their specific expression in different conditions (i.e., tissue, developmental stage, or pathological status) [130]. On the other hand, studies based on a proteomic approach [131,132] are limited by the impossibility of assessing both the activity of the changed proteins and the rate-limiting step of any biochemical pathway. Thus, having resorted to a kinetic approach in which the rate of supply of ATP produced in OXPHOS was continuously measured in MLV, in a first step in this study, we observed that such a rate is dependent on the rate of ADP/ATP exchange via ANT and significantly decreases in spontaneously hypertensive rats (SHRs) as a result of change in both carrier activity and ADP affinity for ANT.

Successively, the ANT rate proved to change in an opposite manner in aging: It decreased in MLV isolated from normotensive Wistar Kyoto rats (WKYRs), but increased in MLV from SHRs due to changes in the ANT function depending on the lifespan, as kinetically shown [133,134,135].

Interestingly, this occurred in two opposite manners: the rate of ATP cytosolic supply decreased in the WKYR, but it increased in the SHRs. Indeed, in aging WKYRs we found a reduced ANT activity in accordance with the altered ANT activity and expression already observed in a variety of human heart failure [136,137]; contrarily, in SHRs the rate of ATP supply outside mitochondria in vitro increased by 300–600% in the physiological ADP concentration range (20–60 μM) [138].

The explanation of the mechanism by which the ANT alteration takes place remains a matter of speculation: the generation of ROS in the electron flow along the mRC [139] and/or the occurrence of reactive aldehydes, already reported to deactivate mitochondrial enzymes, including the ANT [139], were taken into consideration.

ANT damage could also be dependent on a shift in ANT isoform composition, as in [140], as well as on a specific age-dependent decrease in cardiolipin content [141,142].

At present we do not know which is the mechanism by which the Vmax of ANT decreases; however, in light of data in which no difference in ANT expression was found in MLV from four- and eight-month-old rats [137], we assume that a change in the ANT catalytic efficiency occurs in aging.

Our conclusion is not unique: A change in ANT conformation in aging has already been shown via electron microscopy [143], but this paper has the added dimension of showing how such a change relates to ANT function in aging.

Consistently, in transgenic rats overexpressing ANT1 in the heart and crossed with renin-overexpressing rats suffering from hypertension-induced cardiac insufficiency, [137] showed that an accelerated ATP/ADP transport across the mitochondrial membrane improves mitochondrial structure and function, leading to a reduction in fibrosis and an improvement in cardiac tissue architecture, and thus may be a basic principle for new strategies in treating heart disease. In this study, the authors measured the ANT-mediated ADP/ATP exchange rate using ATP D.S., but unlike our original procedure, they preferred to use ATP-loaded mitochondria and then cause ATP efflux following the addition of ADP. What a pity! In fact, it follows that in order for mitochondria to accumulate ATP not only must their metabolism be blocked, but the neo-synthesis of ATP from ADP that has entered the mitochondria must also be blocked. We are facing a situation far from the physiological one!

#### 4.2.4. Apoptosis and Necrosis: ANT Role and Functionality in the Dying Neuron

To describe the specific mechanisms of cell death occurring during neurodegenerative disorders such as Alzheimer’s disease (AD), many investigations, both in vivo and in vitro, have attempted to label the particular pathway of cell death either as apoptosis or as necrosis. Both apoptotic and necrotic cells can be found in AD tissue, and both death types can overlap, sequentially occur under certain conditions, and not be detected unequivocally.

Therefore, although many in vivo and in vitro studies favor apoptosis in AD, there is considerable evidence that a mixture of both events may contribute to neurodegeneration in AD and to its final pathology.

Even though the deposition of β-amyloid and the formation of neurofibrillary tangles due to Tau protein hyperphosphorylation and deposition represent morphological hallmarks of the disease, several impressive lines of evidence suggest that both lesions are not sufficient to cause the AD neurodegenerative process.

Starting from this awareness, with the aim of shedding light on the molecular mechanisms responsible for the onset of AD, for about 30 years our study essentially has made use of an AD cell model, which consists of cerebellar granule cells (CGCs) undergoing apoptosis due to potassium deprivation, but we also used the characteristic AD hallmarks, in particular the smaller and more potent NH_2_26–44-tau peptide combined or not with Aβ 1-42, the major pathogenic species in AD [144], for our ex vivo experiments on mitochondria function, observing that they cooperate by potentiating the ANT-1 impairment ex vivo in coupled neuronal mitochondria (see below).

Deliberately leaving aside the studies carried out on events that progressively cause death, here we report only those studies in which ANT has been studied, highlighting how its role and functionality in the dying cell vary (Figure 3).

##### 4.2.4.1. The Dual Role of the Mitochondrial ANT in Apoptosis

One of the outstanding problems about the role of mitochondria in apoptosis concerns ATP production and utilization in cytosol, where ATP is used in a variety of processes, including caspase activation [154]. Thus, the role of ANT in apoptosis is crucial. Although the mPTP is suggested to be an important mediator of apoptosis (see Section 3.6.2), we measured ADP/ATP exchange and mitochondrial permeability transition (mPT) as a function of time after the induction of apoptosis and we concluded that the function of ANT in cells en route to apoptosis is as follows: In early apoptosis a decrease in the transport efficiency of ANT occurs, caused by a ROS-mediated post-translational modification as shown by its prevention by superoxide dismutase (SOD) and resulting in reduced affinity for ADP. This finding was concomitant with that of Nakagawa [155], who suggested that impairment of ANT is due to ROS-dependent cardiolipin oxidation. In late apoptosis, together with cyclosporin A-sensitive mPTP opening, a further progressive decrease in ANT transport function, i.e., of ADP/ATP exchange, which can be prevented by the caspase inhibitor z-VAD, occurred. Consistently, Xia et al. [156] found that recombinant caspase-3 directly induced mPTP opening and the associated mitochondrial apoptotic phenotype.

It is clear that in this experimental context, ATP D.S. was well suited to measuring the rate of ATP production, obviously reflecting the mitochondrial impairment that occurs during apoptosis [157], which is fitting given that the system measures the exchange in conditions close to real. However, the usefulness of the type of experiments carried out lay precisely in verifying from time to time that the ultimate effect of advancing apoptosis-dependent cellular deterioration as well as that operated by the molecules used, i.e., SOD and z-VAD, safeguard the limiting step of the global process, i.e., the ADP/ATP exchange.

We believe that in this context it is appropriate to dwell on an elegant experiment in which both ADP/ATP exchange and mPTP opening, measured as ADK release into the extramitochondrial phase, were investigated as a function of time (0–8 h) of apoptosis [145]. In early apoptosis (up to 3 h), the rate of ADP/ATP exchange was found to decrease with respect to the control, in which it remained constant at all times, with inhibition increasing with the progression of apoptosis; in the same time range, no or negligible mPTP opening was found. In late apoptosis, a further decrease in ADP/ATP exchange was found, whereas mPTP opening occurred over the 3–8 h time range in apoptotic cells, showing that during apoptosis ANT functionality gives way to mPTP opening.

In order to further refine this type of experiment and to obtain some insight into the mechanism by which these processes occur as well as to determine whether the two processes are really linked to one another, we treated cells undergoing apoptosis with a variety of compounds designed to inhibit certain activities that participate in the processes leading to apoptosis.

Then, in cells incubated with SOD, i.e., under No-ROS conditions, almost complete prevention of ANT impairment was found in early apoptosis, showing that the decrease in ANT transport efficiency depends on ROS production. In late apoptosis, in spite of the negligible change in ROS production [145], progressive impairment in ANT transport was found but to a lesser extent than with apoptotic cells. No mPTP opening was found in early apoptosis, showing that ROS cannot themselves cause the mPT, which occurred only in the 3–8-h time range, but to a lesser extent than the control, i.e., in the absence of SOD.

Interestingly, in apoptotic cells treated with z-VAD, a molecule that prevents caspase activity, no opening of the mPTP occurred either in early or late apoptosis, showing conclusively that mPTP opening requires caspase action. Furthermore, the time-dependent decrease in ANT activity in early apoptosis paralleled that found in control cells, but in late apoptosis no significant further reduction was found, showing that only the late ANT impairment is caspase dependent. As expected in light of the above results, when both ROS and caspase were put out of action, i.e., in the presence of SOD and z-VAD added simultaneously, ANT activity remained largely unaffected with respect to normal cells, and no mPT occurred at all [145].

In order to summarize the data, ANT is damaged in two phases as far as ADP/ATP exchange is concerned, with initial damage due to ROS and further additional damage owing to caspase. Conversely, even if damaged by ROS, ANT can still translocate adenine nucleotides, but cannot participate in mPTP formation. Later, as a result of caspase action, ANT begins to participate in mPTP opening.

Furthermore, the application of a mathematical model showed that loss of ANT transport function and mPTP opening are inversely correlated in late but not in early apoptosis. Namely, the loss of ANT transport function and onset of mPTP are independent during early apoptosis, but the situation changes over the period of 3–5 h, when they proceed in parallel. That is, in the 5–8-h phase of apoptosis, the progressive decrease in the ANT-dependent transport function occurs simultaneously with mPTP opening.

That a metabolic shift from oxidative phosphorylation to glycolysis (i.e., the Warburg effect) occurs in AD accompanied by an increase of both activity and level of HK-I was discovered some years later [146]. These findings proved that in the early phase of apoptosis VDAC1 activity progressively decreases in concomitance with its physical interaction with HK-I. Instead, in the late phase of apoptosis, compelling evidence suggests that glucose-6-phosphate disrupts the physical interaction between the two proteins, then determining re-opening of the channel and the recovery of VDAC1 function, resulting in a reawakening of the mitochondrial function—which, as reported in [146], was apparently numb, thus inevitably leading to cell death [146].

This study was particularly laborious because for the first time ANT1 and VDAC1 activities were estimated in the same experiment by following the ADP/ATP exchange across mitochondrial membranes measured as in [145], in the absence or in the presence of either ATR or 4,40-diisothiocyanostilbene-2,20-disulfonic acid (DIDS), i.e., cell permeable blockers of ANT and VDAC1 [147,158,159], respectively.

As usual, in a typical experiment, cell homogenate was treated with Ap5A, thus preventing mitochondrial ATP synthesis in a manner not dependent on oxidative phosphorylation, and then incubated in the presence of the ATP detecting system. As a result of ADP addition, the appearance of ATP in the extramitochondrial phase was observed as an increase in absorbance due to the formation of NADPH due to (i) ADP uptake into mitochondria in exchange for endogenous ATP, (ii) ATP synthesis from imported ADP via ATP synthase, and (iii) efflux of the newly synthesized ATP from the mitochondria in exchange for further ADP. Up to this point everything went according to the common experimental fees already adopted to measure ADP/ATP exchange.

The decrease in NADPH formation occurring in the presence of DIDS or ATR showed that (i) the exchange of ADPext with ATPint exclusively occurs through the outer and inner mitochondrial membranes, respectively; (ii) both the membranes are intact; and (iii) the exchange is mediated by protein/s. Therefore, as already done in the previous work in order to establish the limiting step of the global process, applying the control-strength criterion and using DIDS at different concentrations, we observed that the rate of absorbance increase, i.e., ADP/ATP exchange, in the absence of an inhibitor was lower than the value corresponding to the straight line that intercepts the *Y*-axis at zero inhibitor concentration—obtained by interpolating the experimental points of absorbance increase rate in the presence of DIDS—thus showing that (i) the inhibited step of measured processes, i.e., the ADP/ATP exchange across MOM via VDAC1, is not the limiting step and (ii) the reciprocal value of the intercepts to the ordinate axis is a measure of the inhibited process, i.e., VDAC1 activity. In contrast, the intercept of the *Y*-axis in the case of the ADP/ATP exchange, inhibited by ATR, coincides with the rate value in the absence of an inhibitor, according to [145].

That ANT is a target of ROS in apoptosis was further confirmed by the use of certain dietary flavonoids known to exert beneficial effects on the central nervous system and to affect neuronal apoptosis [160,161]. The flavonoids genistein and daidzein, which are present in soy, but not catechin or epicatechin, which are present in cocoa, all used in a dietary blood concentration range, were found to prevent apoptosis in a manner consistent with their antioxidant activity [148], as depicted in Figure 3. A detailed investigation of the effect of these compounds on certain mitochondrial events that occur in cells en route to apoptosis showed that genistein and daidzein prevented the impairment of glucose oxidation and mitochondrial coupling, reduced cytochrome c release, and prevented both impairment of ANT and opening of the mitochondrial permeability transition pore, strongly suggesting that the prevention of apoptosis depends mainly on the antioxidant properties.

Furthermore, and for the first time, we demonstrated that extracellular ADP prevents the impairment of mitochondrial ANT-1 [147]. ADP, released into the extracellular space in the brain by multiple mechanisms, can interact with its receptor or be converted, through the actions of ectoenzymes, to adenosine.

ADP, released into the extracellular space in the brain by multiple mechanisms, can interact with its receptor or be converted, through the actions of ectoenzymes, to adenosine, which is subsequently taken up via the ADO transporter [147]. Consistent with this hypothesis, the internalization process is sensitive to the antagonist of purine receptor and the inhibitors of ectonucleotidase and nucleoside transporter [147], as depicted in Figure 3.

Our findings demonstrate that extracellular ADP inhibits the proapoptotic stimulus supposedly via (i) inhibition of ROS production during early stages of apoptosis, an effect mediated by its interaction with cell receptor/s, a conclusion validated by the increase in cell AOX system; and (ii) safeguarding of the functionality of the mitochondrial adenine nucleotide-1 translocator (ANT-1), an effect mediated by its plausible internalization into cells occurring as such or after its hydrolysis by means of plasma membrane nucleotide metabolizing enzymes, and resynthesis into the cell.

Surprisingly, in this paper we showed that, although ADP is essential to completely protect mitochondrial ANT-1, its effect on mPTP opening is only partial, thus suggesting that ANT-1 involvement is not the primary event leading to mitochondrial permeabilization and then to cell death, consistent with [145]. Furthermore, the complete prevention of death exerted by ADP but the partial prevention of mPTP opening confirmed that death and mPTP are not strictly dependent on each other in apoptosis.

Shifting the focus towards the effect of ATR on viability, this selective ANT inhibitor prevents ADP protection, thus suggesting that the molecular mechanism responsible for the ADP protective effect is not unique. Since complete prevention of ADP protection occurred only when ATR and pyridoxalphosphate-6-azophenyl-2′,4′-disulfonic acid, a functionally selective antagonist of purine receptor, were added together, it was inevitable to assume that the ADP protective effect on cell death is mediated both by the activation of the AOX system, occurring in cells undergoing apoptosis treated with ADP, and by its direct interaction with the mitochondrial carrier that drives the ADP/ATP exchange. Nevertheless, it should also be considered that the activation of AOX enzymes results in depressed levels of ROS, which in turn reduces the ANT-1 activity, as in [145].

In the same paper we observed that extracellular ADP also protects ANT-1 from the toxic action of the two AD peptides, i.e., Aβ1–42 andNH_2_htau, further corroborating the molecular mechanism of neuroprotection by ADP. ADP’s capability to protect the carrier from toxic NH_2_htau can be easily explained when considering that the inhibition exerted by this fragment on ANT-1 is competitive, that is, it interacts with the catalytic site to which the substrate binds.

On the assumption that neuropathies are the result of neuronal apoptosis, the identification of compounds that, like ADP, are able to protect neurons against apoptosis is highly desirable.

##### 4.2.4.2. NH_2_htau and Aβ1-42 Peptides Impair Mitochondrial ANT-1 in Alzheimer’s Disease

Studies on several amyloid precursor proteins and tau transgenic mouse models suggested that a possible link between these two characteristic AD hallmarks might be an early mitochondrial dysfunction, particularly at synapses associated with increased oxidative stress [162].

Interestingly, we observed that high intracellular levels of pathogenetic NH_2_htau are able to interfere with the mitochondrial biology leading to a drop in ATP and eventually to neuronal death, as reported in [149,150]. This is a typical example where ANT is the target of a toxic compound produced in the cell affected by AD.

A few years earlier, Amadoro et al. [151,152] showed that high intracellular levels of tau N-terminal fragments lacking the first 25 amino acids evoke a potent neurotoxic effect and induce a necrotic type of cell death, as sustained by protracted stimulation of N-methyl-D-aspartate (NMDA) extrasynaptic receptors. The NH_2_-26–44 tau fragment was the minimal active moiety, which retained a marked necrotic effect. In a meticulous study, we showed that the NH_2_-26-44 tau fragment can impair OXPHOS, as investigated in vitro with homogenates containing intact coupled mitochondria, with this occurring as a result of ANT impairment. Such an impairment could account for the reduced availability of ATP already proposed as the cause of the excitotoxic NMDA receptor activity mediated death in Alzheimer’s disease. In this case, the reduced ATP availability in the cytosol could in turn cause glutamate release from the cell and then excitotoxicity [151,152].

The experimental approach used to address the role of NH_2_-26-44 fragments in neuron energy metabolism highlighted that both cytochrome oxidase and ANT are targets of NH_2_-26-44 tau fragments, but ANT is the unique mitochondrial target responsible for impairment of OXPHOS. The mechanism by which the NH_2_-26-44 tau fragment impairs ANT is at present a matter of speculation. Since Yang and Ksiezak-Reding [163] showed that as a consequence of glycation, paired helical filament-tau from AD and advanced glycation end-tau generate oxygen free radicals and since Cente et al. [164] reported that expression of a human truncated variant form of tau protein leads to the accumulation of ROS, the possibility that ANT impairment derives from ROS–protein interaction as well as from protein–protein interaction has been considered.

In this regard, several studies exist in which different aspects of putative interaction sites, such as hydrophobicity, residue propensities, size, shape, solvent accessibility, and residue pairing preferences, are examined [149]. As far as the interaction of tau peptides and ANT is concerned, a simple comparison of the peptides and the ANT isoform sequences, based on the six-transmembrane model (three-repeat domain structure) proposed by Klingenberg [165], does not reveal any element that could be predictive of a direct interaction, such as cysteine or proline residues [149].

Nonetheless, it must also be considered that since the NH_2_-26-44 tau fragment is a non-competitive inhibitor, it does not interact with the catalytic but rather with some other site of the enzyme, which could distort the enzyme’s structure, thus also affecting the catalytic binding site. In particular, we believed it was important to characterize the mechanism(s) regulating ANT function not only for the purpose of explaining the pathways contributing to necrosis but also to understand the pathogenesis of various taupathies that may result from the altered function of ANT.

All together the results by Amadoro et al. [151,152] proved that caspase(s) play/s a dual pivotal role in apoptosis not only by degradation of endogenous tau, thereby reducing the intracellular endogenous pool of full-length tau available for binding to microtubules, but also by generating the production of the neurotoxic NH_2_-26–44 tau fragment that impairs the mitochondrial ANT and then ATP cellular availability.

The knowledge that mitochondrial dysfunction, with a key role played by ANT impairment, is of central importance in AD opens a window for new therapeutic strategies aiming to preserve/ameliorate mitochondrial function, and represents an exciting challenge for biochemists.

Subsequently, we showed that the NH_2_-26-44 peptide tau fragment preferentially interacts with Aβ peptide(s) in human AD synapses in association with mitochondrial adenine nucleotide translocator-1 (ANT-1) and CyP-D [149]. The two peptides inhibit the ANT1-dependent ADP/ATP exchange in a noncompetitive and competitive manner, respectively, and together further aggravate the mitochondrial dysfunction by exacerbating the ANT-1 impairment and thereby potentiating and amplifying each other’s deleterious effects [166]. The present work provides additional insights on how tau dysfunction directly causes mitochondria impairment at AD synapses by binding ANT-1 alone or in cooperation with a peptide(s). Consistently, an aggravated mitochondrial impairment has been described in eight- and 12-month-old *APP/PS/tau* transgenic mice combining both plaques and tangles, compared with mice overexpressing *tau* or *APP* alone. In this context, the pathological convergence between tau and Ab at AD synaptic mitochondria may help to explain why the diseased A- or tau-modifying strategies have not singularly given promising results and suggest potential new pathway(s) and target(s) [153] for a combined, more efficient therapeutic intervention of early synaptic dysfunction in AD.

Next, we dissected the molecular mechanism by which NH_2_-htau and Aβ1-42 impair mitochondrial ANT1. Since it is known that ANT contains –SH groups, which are both essential for the catalytic activity and preferential targets for ROS attack, we investigated whether ANT1 inhibition is due to an interplay between these two Alzheimer peptides, ROS and ANT1 thiols [167].

To accomplish this purpose, we turned to mersalyl, a reversible alkylating agent of thiol groups that are oriented toward the external hydrophilic phase [168], to selectively block and protect the –SH groups of ANT-1 in a reversible manner. The ability of Aβ1-42 to induce mitochondrial ROS production and that of mersalyl to protect ANT-1 against truncated NH_2_htau, but not Aβ1-42, was observed.

Then, a molecular mechanism in which the pathological Aβ-NH_2_htau interplay on ANT1 in Alzheimer’s neurons involves the thiol redox state of ANT-1 and the Aβ1-42-induced ROS increase was proposed. This result represents an important innovation because it suggests the possibility of using various strategies to protect cells at the mitochondrial level by stabilizing or restoring mitochondrial function or by interfering with the energy metabolism providing a promising tool for treating or preventing AD.

Our finding was consistent with the paper by Atlante et al. [124]: Singlet oxygen, produced by irradiation of photosensitizer-loaded mitochondria, triggers permeability transition pore (PTP) opening, and thiol reagents, in particular mersalyl, prevent PTP opening when mitochondria are irradiated after the addition of mersalyl. This is in agreement with the observation of Atlante et al. [123], according to which ANT is a major target of ROS-dependent photodynamic action (see above, Section 4.2.1).

Paradoxically, Aβ1-42 protects the cell from the inhibitory effect of NH_2_htau on ANT-1. In this regard, we observed that ADP added to cultured neurons also reduced the tau effectiveness in inhibiting ANT-1, like Aβ1-42 does. It might be possible that binding ADP to the carrier could grant protection against thiol oxidation by (*i)* promoting alterations of the ADP/ATP carrier conformation that change the position of its thiol groups [165,167,169], rendering them not accessible to oxidation by ROS, or (*ii)* changing mitochondria from the orthodox to a condensed configuration, an alteration that may protect thiol groups of membrane proteins against oxidation. This aspect is in agreement with the observation that ROS, while readily reacting with and damaging vital cellular structures, among them lipids, DNA, and proteins [168], can also regulate protein function through the modification of specific thiol(s). In recent years, an increasing number of proteins have been identified that are not damaged by oxidative stress conditions but use ROS-mediated thiol modifications to specifically regulate their function [168].

#### 4.2.5. The ANT-Dependent Export of ATP Changes in Plant-Programmed Cell Death (PCD)

In the transition from the nerve cell, in which apoptosis is responsible for the advancement of neurodegeneration and death in the most famous diseases of the brain, first of all Alzheimer’s, to the plant cell, programmed cell death (PCD) is considered an integral part of the life cycle, being a major component of normal development, preservation of tissue homeostasis, and elimination of damaged cells in response to various forms of abiotic and biotic stresses [170].

The pivotal involvement of mitochondria in the execution of PCD is also undiscussed in plants [171], with characteristics observed in mammalian systems. In an original study carried out by Valenti et al., [172], with the aim of investigating whether and how mitochondria undergo changes in plant PCD, non-photosynthetic Tobacco Bright Yellow 2 (TBY-2) cells were used. In coupled mitochondria isolated from TBY-2 cells undergoing PCD as a result of heat shock, whether and how the ANT-dependent export of ATP synthesized via OXPHOS—and also the ATP production via ADK and the activity of the nucleoside diphosphate kinase (NDPK)—can change during the early phases of PCD in the TBY-2 cell line was investigated. A strong inhibition in the rate of ADP/ATP exchange via ANT with a drastic reduction in transport efficiency in spite of only a minor decrease in the amount of protein was found, suggesting that the overall effect is due essentially to a double impairment of ANT function, i.e., a minor decrease in the protein level and a strong decrease in the catalytic efficiency.

#### 4.2.6. ANT Deficit in Down Syndrome and in Cystic Fibrosis

Mitochondrial dysfunction critically impairs nervous system development and is potentially involved in the pathogenesis of various neurodevelopmental disorders, including Down syndrome, the most common genetic cause of intellectual disability. Therefore, a punctual and functional study was made on certain individual steps involved in mitochondrial ATP production in human skin fibroblasts with chromosome 21 trisomy (DS-HSF), providing new insight into the molecular basis for mitochondrial dysfunction in DS [173].

In the experiments carried out in this study, already the reduced efficiency of externally added ADP in stimulating the rate of oxygen consumption in DS-HSF mitochondria was an indicator of altered OXPHOS capacity. It was therefore necessary to make use of spectroscopic measurements that allowed for the continuous monitoring of ATP synthesized by mitochondria and exported outside mitochondria arising from externally added ADP under conditions in which OXPHOS really occurs [104].

Once again, we availed ourselves of the ATP D.S. in order to investigate further the mitochondrial ATP-related energy production in DS-HSF. Since in mitochondria ATP efflux can occur with the contribution of both ATP synthase/ANT activities (in OXPHOS) and the intermembrane enzyme ADK (which itself, catalyzing the reversible high-energy phosphoryl transfer reaction between adenine nucleotides, can provide ATP from externally added ADP), in this case the authors also used experimental conditions designed to monitor the two pathways separately.

Surely one of the advantages of the ATP detecting method is being able to differentiate the origin of the newly synthesized ATP. In order to selectively measure ATP synthesized via OXPHOS, samples were treated with Ap5A to inhibit ADK [105]. When ADK-related ATP synthesis was selectively monitored, mitochondria were treated with CAT plus OLIGO to block OXPHOS.

DS cells showed an impairment of OXPHOS capacity involving ATP synthase, ANT, and ADK deficit, events that might be ascribed to post-translational modification: the deficit of ANT as well as of ADK activity in DS fibroblasts occurred despite both proteins being not only not differentially expressed, but also up-regulated at the protein level. Interestingly, exposure of DS-HSF to dibutyryl-cAMP, a permanent derivative of cAMP, stimulated ANT, ADK, and ATP synthase activities, whereas H89, a specific PKA (protein kinase A) inhibitor, suppressed this cAMP-dependent activation, indicating an involvement of the cAMP/PKA-mediated signaling pathway in the ATP synthase, ANT, and ADK deficit [173].

In a later study, the same authors showed that both epigallocatechin-3-gallate and resveratrol, natural polyphenols, reverse the severe impairment of mitochondrial bioenergetics and biogenesis in neural progenitor cells isolated from the hippocampus of Ts65Dn mice, a widely used model of DS that recapitulates many major brain structural and functional phenotypes of the syndrome, including impaired hippocampal neurogenesis [174]. The rescuing of the in vitro impaired neurogenesis, likely linked to the activation of the PGC-1α/Sirt1/AMPK axis, suggested that both EGCG and resveratrol exert a potential beneficial action for treatment in DS.

Recently, in contrast to what is the hottest research topic in cystic fibrosis (CF) disease, essentially focused on the cystic fibrosis transmembrane regulator (CFTR) protein [175,176], we first launched into a study essentially scouting mitochondrial bioenergetics, which was extremely necessary and essential to provide the groundwork for future research aimed at understanding the molecular mechanisms responsible for the involvement of mitochondria in CF and to identify the proteins primarily responsible for the F508del-CFTR-dependent mitochondrial impairment and thus reveal potential novel targets for CF therapy.

Hence, in the context of the research called ‘”Relationship between mitochondria and F508del-CFTR in Cystic Fibrosis”—supported by the Italian Cystic Fibrosis Research Foundation FFC#1/2015 Project to A.A.—with the aim of studying the role of mitochondria related to the deficiency of F508del-CFTR function, i.e., loss of phenylalanine residue at position 508, responsible for defective transport of chloride ions across airway epithelial tissues [177,178,179,180], we characterized mitochondrial function, in particular as it regards the steps of OXPHOS and ROS production, in airway cells either homozygous for the F508del-*CFTR* allele or stably expressing wt-*CFTR*.

We approached this issue essentially by using cell homogenate as a model system, as it contains intact and coupled mitochondria and provides a “physiological” environment for the investigation of the participation of mitochondria in cellular functions, thus allowing for simultaneous monitoring of the closely interlinked cellular and mitochondrial machineries. All the steps leading to mitochondrial ATP synthesis and to its export into the cytoplasm via ANT were examined one by one [181]. Consistently, an important note is that as ANT-dependent ATP availability in the cytosol regulates CFTR channel activity gating [182], it follows that the activity of the ADP/ATP translocator, in concert with cytosolic enzymes that consume ATP, is really critical in CF by playing a decisive role in the maintenance of cytosolic ATP levels.

Once again, also in this study, the procedure [104,145] that allows for the continuous monitoring of ATP efflux from mitochondria in the cell homogenate incubated with ADP appeared fit for the purpose of investigating ANT activity once we made sure that the rate of ATP efflux mirrored the rate of ADP/ATP exchange, which was verified by applying control flux analysis, as in [145]. We found that, together with other steps of OXPHOS, ANT-dependent ADP/ATP exchange was impaired in CF cells and, importantly, treatment of CF cells with the small molecules VX-809 and 4,6,4′-trimethylangelicin, which act as “correctors” for F508del CFTR by rescuing the F508del CFTR-dependent chloride secretion, significantly improved the mitochondrial parameters towards values found in the airway cells expressing wt-*CFTR*.

We cannot currently provide any molecular mechanism underlying how CFTR dysfunction affects so many parameters of mitochondrial function, and in particular on ANT activity, nor how corrector-induced increased expression of cell surface *CFTR* is able to repair these mitochondrial dysfunctions. However, the data per se, beyond strongly suggesting that the restorative action provided by the correctors in CF cells is linked to the rescue of chloride channel activity, corroborate the validity of the method by which ANT activity was measured since the small corrector molecules have no effect per se on mitochondrial function in control-CFTR cells.

## 5. Closing Remarks

In this review we discussed the wide relevance of the ATP detecting system performed to measure the ANT-mediated ADP/ATP exchange by highlighting the essential aspects of its versatile applicability in pathophysiology. We undertook this study, in which literature data alternates with our own stories concerning our personal steps taken in the scientific world, after “taking a walk” through Google Scholar and realizing that all the research concerning ADP/ATP exchange as well as the synthesis of ATP via OXPHOS, measured under conditions as close as possible to the real ones, i.e., by using the ATP D.S., had received less visibility, i.e., fewer citations, than it deserves given the importance of the topic.

This was the reason that prompted us to group all the studies reported above in an attempt to create a sort of common thread that goes from the very first experimental attempt to continuously measure the OXPHOS process of mitochondrial ATP synthesis, through all the uses that have been made of this method—remaining still “new” for those who do not yet know it—up to its current application in a more complex study that saw cell systems and/or tissues carrying genetic and/or induced diseases as a playing field through the use of specific effectors.

We hope that it will be of help to researchers who want to adopt it in the future.

## Figures and Tables

**Figure 1 ijms-22-04164-f001:**
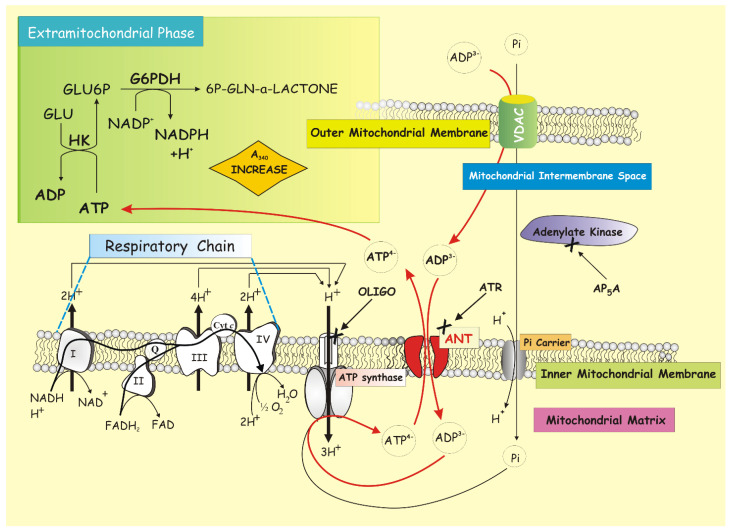
Functional activity of ANT-mediated ADP/ATP exchange revealed by an ATP detecting system. The ATP detecting system (ATP D.S.), depicted in the upper left panel, reveals the appearance of ATP in the extramitochondrial phase following the addition of ADP to the mitochondria and the synthesis of ATP via OXPHOS. As shown in the picture, since the activity of the ANT-dependent electrogenic ADP/ATP exchange strictly depends on the mitochondrial membrane potential, mitochondria must be actively “respiring and phosphorylating,” i.e., the activity of mRC that generates the electrochemical gradient transmembrane, and ATP synthesis must be closely coupled for the exchange to occur.

**Figure 2 ijms-22-04164-f002:**
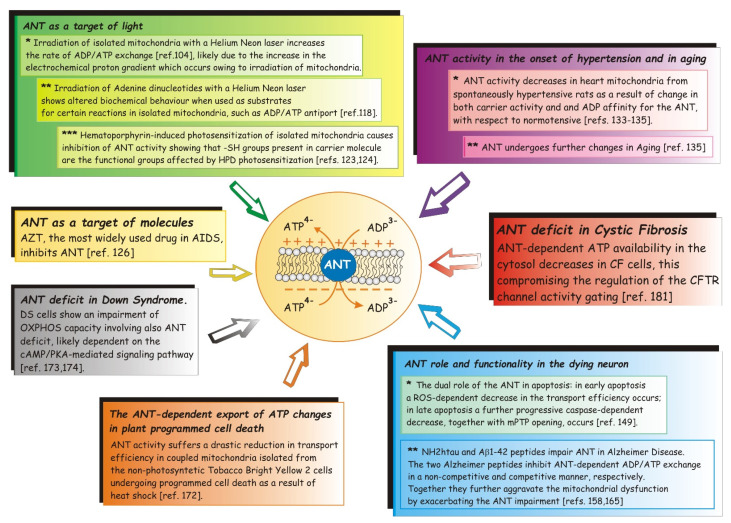
ANT is the target of light or molecules or even of the two Alzheimer’s proteins and its activity is impaired under various pathological conditions affecting the host cell.

**Figure 3 ijms-22-04164-f003:**
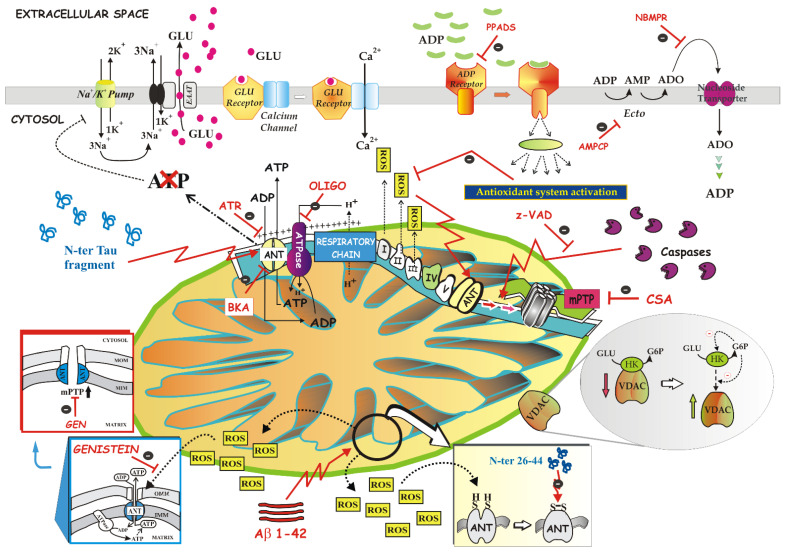
The picture collects data from several studies on the role of ANT and its functionality in the neurons sentenced to death for apoptosis or necrosis. In particular, the dual role of ANT, harboring two opposite functions, i.e., a vital function important to ensuring the life of the cell, and the other lethal—such as protein involved in the participation of mPTP opening—leading the cell to death, is depicted. The effect of NH_2_htau and Aβ1-42 peptides impairing mitochondrial ANT-1 in Alzheimer’s disease is also shown. The data represented here refer to the reference numbers [145,146,147,148,149,150,151,152,153].

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
