# Peer review of "A Walk in the Memory, from the First Functional Approach up to Its Regulatory Role of Mitochondrial Bioenergetic Flow in Health and Disease: Focus on the Adenine Nucleotide Translocator"

_ijms, 2021, doi:10.3390/ijms22084164_

Round 1

Reviewer 1 Report

This review is an amazing journey in a time machine, rather than a typical review. Which is why, I am wondering if it should be published in IJMS or rather should be a book chapter in the handbook. Despite this concern, which will be later on decided by the Editors, I was absolutely charmed by the combination of the raw scientific knowledge with the stories about Author’s straggling the mitochondrial studies. This story engulfed my attention completely, and I read it in one go. Each practical story memorized my first experiences with biochemical assays. Unbelievably, even if my lab work has started only slightly more 10 years ago, I was facing exactly the same problems with biochemical studies. This Review sounds like an opening speech given by the Professor to young researchers, maybe somewhere at the opening of the academic year or international conference or visiting lecture. All these events, which due to the epidemic limitation, will not take place anytime soon. Therefore, by this Review, the Authors gave a chance to young Researchers starting in biochemistry to understand the method limitations, restrictions and the quality control rules. Therefore, even if (as mentioned) maybe this Manuscript is not a typical review, I am strongly recommending this work for further publishing processes.

Minor:

Small grammar slips, e.g:

Line 11 “…role of gate-keeper of cellular energy…”

Line 77 “… hat energy production…”

Line 90 “…fulfil…”

Headlines: some of them were wrote in italic, the other remained normal

Half abstract was written with one writing style, while the second part with Time New Roman.

Thus, I am recommending a copyediting check-in of this MS.

Reviewer 2 Report

The language is too colloquial to be published in a scientific journal, it contains many grammatical errors and the phrases are too extended.

The manuscript is written too descriptively which makes it very difficult to focus on the data reviewer. Major editing is necessary to help understand the work and enhance the impact of data.

The figures are not in an appropriate format for a scientific journal, these should be with a formal style, not a figures for a slides or posters.

Round 2

Reviewer 2 Report

The language is too colloquial to be published in a scientific journal, it contains many grammatical errors and the phrases are too extended. The manuscript is written too descriptively which makes it very difficult to focus on the data reviewer. Major editing is necessary to help understand the work and enhance the impact of data. The figures are not in an appropriate format for a scientific journal, these should be with a formal style, not a figures for a slides or posters.

Response to the Reviewer

We thank the Reviewer for the careful reading of the Review.

As he/she points out, there were grammatical errors and the phrases were too extended. We have tracked down the errors and remodelled too long sentences. The corrections made are visible in the revised version.

We are saddened that the Reviewer has encountered difficulties to focus on the data. The authors, before starting to write, thought a lot about the presentation of this Review. It would have been easy and obvious to present the data -proposed and revisited -according to the canons of a 'typical review', but certainly the outcome would have been aseptic and limited to a list of studies performed. Our intention was precisely to combine the scientific data of the literature with the stories about the mitochondrial studies of the Authors. This allowed us to give a chance to Researchers and / or interested Readers to understand the method limitations, restrictions and the quality control rules that this dusted off method presents. From this derives the 'language too colloquial' -by which the Reviewer was struck -which, however, we do not believe compromises the scientific content of the Review.

About Figures, since the pivotal topic of the Review concerns the applicability of a METHOD, perfected about 30 years ago, we believe that no figure in an appropriate format for a scientific journal can describe it. However, as has happened several times when writing Review articles (and also original articles!) for Journal of MDPI family, the policy of these journals is to promote the graphical representation of article contents, as a guide to reading for the Researcher. Therefore, all 3 figures (all original pictures!) -intend to valorise the studies carried out using the ATP detecting system. In addition to Fig 1 which shows how the ATP detecting system method measures the activity of the ANT (and also the activity of VDAC in a same experiment, an otherwise impossible undertaking !!), Fig. 2 summarizes all the studies carried out, made possible thanks to the potential resources of the ATP D.S. and lastly, Fig. 3 -to whose creation care and attention to the smallest details have been applied -collects various experimental fields -all about the Alzheimer's disease study -in which the ANT, studied with the method in question, occupies a leading position for the logic and strategy used to investigate some aspects of the two AD proteins, tau and beta-amyloid. The use of colour, which is perhaps more applied to the preparation of slides and / or Posters ­with the intent of attracting the attention of the reader and / or congressman -has perfectly the same intent: to attract the attention of the interested reader. Nevertheless, it should be taken into account that finally these Journals of MPDI family allow, unlike other canonical journals, the inclusion of this type of figures in the articles.

The authors have substantially improved the manuscript. I am still thinking that figures should have a more standard format but if the MPDI group accepts them I have no more objection.